# SELF-SUPERVISED REGRESSION LEARNING USING DOMAIN KNOWLEDGE: APPLICATIONS TO IMPROVING SELF-SUPERVISED IMAGE DENOISING

## ABSTRACT

Regression that predicts continuous quantity is a central part of applications using computational imaging and computer vision technologies. Yet, studying and understanding self-supervised learning for regression tasks – except for a particular regression task, image denoising – have lagged behind. This paper proposes a general self-supervised regression learning (SSRL) framework that enables learning regression neural networks with only input data (but without ground-truth target data), by using a designable operator that encapsulates domain knowledge of a specific application. The paper underlines the importance of domain knowledge by showing that under some mild conditions, the better designable operator is used, the proposed SSRL loss becomes closer to ordinary supervised learning loss. Numerical experiments for camera image denoising and low-dose computational tomography denoising demonstrate that proposed SSRL significantly improves the denoising quality over several existing self-supervised denoising methods.

## 1 INTRODUCTION

Deep regression neural network (NN)-based methods that can accurately predict real- or complex-valued output have been rapidly gaining popularity in a wide range of computational imaging and computer vision applications including image denoising (Vincent et al., 2010; Xie et al., 2012; Zhang et al., 2017), image deblurring (Xu et al., 2014), image super-resolution (Dong et al., 2016; Kim et al., 2016), light-field reconstruction (Chun et al., 2020; Huang et al., 2020), object localization (Szegedy et al., 2013), end-to-end autonomous driving (Bojarski et al., 2016). Yet, they lack a general self-supervised learning framework.

In training a regression NN $f : \mathbb{R}^N \to \mathbb{R}^M$, the most prevalent supervised learning approach minimizes the mean square error (MSE) between what $f$ predicts from an input $x \in \mathbb{R}^N$ and a ground-truth target $y \in \mathbb{R}^M$:

$$\min_f \mathbb{E}_{x,y} \|f(x) - y\|_2^2. \tag{1}$$

Learning a denoising or refining NN uses (1) with $M = N$ – dubbed Noise2True – where $x$ is a corrupted image and $y$ is a clean (i.e., ground-truth) image. However, it is challenging or even impossible to collect many clean images $y$ in many practical applications, motivating research on self-supervised learning for image denoising (Ulyanov et al., 2018; Soltanayev & Chun, 2018; Krull et al., 2019; Batson & Royer, 2019; Laine et al., 2019; Moran et al., 2020; Quan et al., 2020; Xu et al., 2020; Hendriksen et al., 2020; Xie et al., 2020; Huang et al., 2021) – called self-supervised image denoising. To learn a denoiser $f$ with single noisy images, a popular self-supervised image denoising method, Noise2Self (Batson & Royer, 2019) (see also the concurrent work (Krull et al., 2019)), and its sophisticated relaxation, Noise2Same (Xie et al., 2020), study the following MSE minimization problem:

$$\min_f \mathbb{E}_x \|f(x) - x\|_2^2. \tag{2}$$

These methods use some partitioning schemes in (2) to avoid that its optimal solution is just the identity mapping $\mathcal{I}$. Noise2Noise (Lehtinen et al., 2018) that learns a denoiser with pairs of two

---

This article has appendix and supplement. The appendix and supplement number sections, figures, and tables with the prefix "A" and "S", respectively.

independent noisy images, is a pioneer work for self-supervised image denoising. Motivated by Noise2Noise, several self-supervised image denoising methods such as Noise2Inverse (Hendriksen et al., 2020), Neighbor2Neighbor (Huang et al., 2021), and (Soltanayev & Chun, 2018; Moran et al., 2020; Xu et al., 2020) emulate pairs of two independent noisy images, by applying partitioning or adding simulated noise to single noisy measurements. All the aforementioned methods have been developed based on some noise assumptions including pixel-wise independent noise (Krull et al., 2019; Xie et al., 2020; Huang et al., 2021) and zero-mean noise (Lehtinen et al., 2018; Batson & Royer, 2019; Xie et al., 2020). Yet, they lack design flexibility that might relax such noise assumptions and further improve the denoising performance of NNs. Some works estimate statistical parameters of noise, such as noise histogram (Krull et al., 2020) and parameters of Gaussian mixture noise model (Prakash et al., 2021).

This paper presents new insights on this topic. The paper proposes a general self-supervised learning framework for regression problems, which we refer to as self-supervised regression learning (SSRL). Proposed SSRL enables learning regression NNs with only input samples, by using a designable operator that can encapsulate domain knowledge of a specific application. Our main results show that under some mild conditions (e.g., in image denoising, statistical noise properties in $x$), the better desinable operator is used, the proposed SSRL loss becomes closer to ordinary supervised learning loss. In addition, a designable operator with good domain knowledge can relax noise assumptions of existing self-supervised denoising methods. Numerical experiments for camera image and low-dose computational tomography (CT) denoising with both simulated and real-world datasets – corrupted by only single noise realization – demonstrate that the proposed SSRL framework significantly improves denoising quality compared to several existing self-supervised denoising methods. Put together, our findings provide new insights into how using good domain knowledge can improve self-supervised denoising, underscoring the benefits of understanding application-specific knowledge in SSRL. (Section S.1 further elaborates the contributions of the paper.)

## 2 SSRL USING DOMAIN KNOWLEDGE

The proposed SSRL loss is given by

$$\mathbb{E}_x \| f(x) - g(x) \|_2^2, \tag{3}$$

where $g : \mathbb{R}^N \to \mathbb{R}^M$ is a designable operator encapsulating domain knowledge of a specific application. We will incorporate some sophisticated setups in (3) such that $f$ obtained by minimizing (3) cannot merely be $g$. Although related theorems (see later) hold for any $M$, we mainly focus on practical image denoising applications with pseudo-target $g(x) \not\approx y$.

### 2.1 MOTIVATION

This section empirically shows that understanding domain knowledge is important for designing $g$ in the proposed SSRL loss (3). The following camera image denoising examples demonstrate that well-designed $g$ with good domain knowledge improves the denoising performance of learned $f$ via (3).

Suppose that camera images are corrupted by salt-and-pepper noise. Consider two example setups for $g(\cdot)$, median filtering and BM3D denoiser (Mäkinen et al., 2020), denoted by median($\cdot$) and BM3D($\cdot$), respectively. Figure 1 compares the denoising performance of minimum $f^\star$ with the two aforementioned $g$ setups: $f^\star$ with median filtering significantly im-

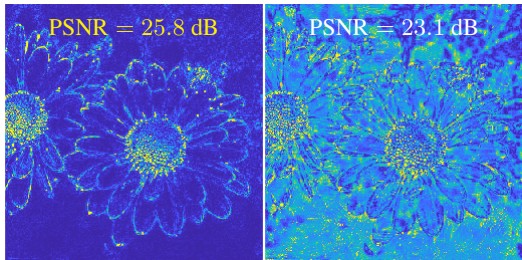

Figure 1: Error map comparisons of denoised images from (3) using $g(x) = \text{median}(x)$ (**left**) and $g(x) = \text{BM3D}(x)$ (**right**) (blue and yellow denote 0 and 50 absolute errors, respectively). Peak signal-to-noise ratio (PSNR) values are averaged.

proved that with BM3D denoiser. This is not surprising, as median filtering is widely known to be effective in reducing salt-and-pepper noise (Bovik, 2010, §3.2). This result emphasizes the importance of understanding domain knowledge of specific applications in proposed SSRL.

## 2.2 PRELIMINARIES

We first introduce the $\mathcal{J}$-complement between two functions $f$ and $g$:

**Definition 1.** *For a given partition $\mathcal{J} = \{J_1, \ldots, J_B\}$ ($|J_1| + \ldots + |J_B| = N$) of the dimensions of input $x \in \mathbb{R}^N$, functions $f : \mathbb{R}^N \to \mathbb{R}^M$ and $g : \mathbb{R}^N \to \mathbb{R}^M$ are called $\mathcal{J}$-complementary, if $f(x_{J^c})$ does not depend on $g(x_J)$ for all $J \in \mathcal{J}$, where $J^c$ denotes the complement of $J$, and $(\cdot)_J$ denotes a vector restricted to $J$.*

That is, $f$ and $g$ use information from outside and inside of $J$ to predict output and give pseudo-target, respectively. In denoiser learning (where $M = N$), Definition 1 specializes to the $\mathcal{J}$-invariance of $f$ (Batson & Royer, 2019), by setting $g = \mathcal{I}$. Incorporating Definition 1 into the SSRL loss (3) is a straightforward approach to avoid that optimal $f$ is just $g$ in (3). The proposed SSRL framework assumes the followings:

*Assumption 1)* $x_J$ and $x_{J^c}$ are conditionally independent given $y$, i.e., $p(x|y) = p(x_J|y)p(x_{J^c}|y)$.
*Assumption 2)* $\mathbb{E}[g(x)|y] = y$.
*Assumption 3)* $f$ and $g$ are (Borel-)measurable.

In denoiser learning, Assumption 1 holds if noise in each subset $J \in \mathcal{J}$ is conditionally independent from that in $J^c$, given $y$. Assumption 1 can be satisfied in general regression NN learning, by adding small randomized perturbations (independent of $y$) to either $J$ or $J^c$, similar to Moran et al. (2020). Assumption 2 suggests a direction for designing $g$: suppose that $x$ has non-zero-mean noise; one then can design $g$ to make noise zero-mean using the domain knowledge. Assumption 3 is satisfied if $f$ and $g$ are continuous. This condition is mild because many regression NNs $f$ are continuous – where their modules, convolution, matrix-vector multiplication, rectified linear unit activation, max pooling, etc. are continuous – and one can design $g$ with measurable or continuous function.

Finally, observe that the proposed SSRL loss (3) can be rewritten by

$$\mathbb{E}_x \|f(x) - g(x)\|_2^2 = \mathbb{E}_{x,y} \|f(x) - y\|_2^2 + \|g(x) - y\|_2^2 - 2\langle f(x) - y, g(x) - y \rangle. \qquad (4)$$

We aim to either remove or control the third term, incorporating the $\mathcal{J}$-complement and/or Assumptions 1-3 introduced above.

## 2.3 SSRL USING DOMAIN KNOWLEDGE WITH $\mathcal{J}$-COMPLEMENT

This section studies SSRL loss (3) minimization over $f$ that is $\mathcal{J}$-complementary of $g$. Our first main result shows that under Assumptions 1–3, the SSRL loss (3) with the $\mathcal{J}$-complement is the sum of the ordinary supervised learning loss and variance of $g(x) - y$, i.e., the third term in (4) vanishes.

**Theorem 2.** *Under Assumptions 1–3, the SSRL loss (3) with the $\mathcal{J}$-complement in Definition 1 becomes*

$$\mathbb{E}_x \|f(x) - g(x)\|_2^2 = \mathbb{E}_{x,y} \|f(x) - y\|_2^2 + \|g(x) - y\|_2^2. \qquad (5)$$

*The following equality similarly holds for any $K \in \mathcal{K}$: $\mathbb{E}_x \|f(x)_K - g(x)_K\|_2^2 = \mathbb{E}_{x,y} \|f(x)_K - y_K\|_2^2 + \|g(x)_K - y_K\|_2^2$, where $\mathcal{K}$ is a partition of $\{1, \ldots, M\}$, and $f(\cdot)_K$ and $g(\cdot)_K$ denote $f(\cdot)$ and $g(\cdot)$ restricted to $K$, respectively. The optimal solution for (5) is given by*

$$f^\star(x_{J^c}) = \mathbb{E}[g(x_J)|x_{J^c}] = \mathbb{E}[y|x_{J^c}]. \qquad (6)$$

*Proof.* See Section A.1 in the appendix. □

The result (6) suggests another direction for designing $g$: we aim to design good $g$ that can make $g(x_J)$ close to $y$. Using such $g$, optimal solution of LHS in (6), $\mathbb{E}[g(x_J)|x_{J^c}]$, becomes close to its supervision counterpart, $\mathbb{E}[y|x_{J^c}]$. Consequently, such $g$ reduces the second term $\mathbb{E}_{x,y} \|g(x) - y\|_2^2$ in RHS of (5), leading (3) closer to (1). If $g$ is ideal such that $g(x_J) = y$, the SSRL loss (3) becomes the usual supervised learning loss. In designing $g$, domain knowledge of specific application is crucial. Domain knowledge includes noise properties in $x$ and pre-trained NN by existing self-supervised denoising, such as Noise2Self (Batson & Royer, 2019) and Noise2Noise (Lehtinen et al., 2018).

Specifically, the proposed SSRL loss using the $\mathcal{J}$-complement is given by

$$\mathcal{L}_{\text{ind}}(f) \triangleq \sum_{J \in \mathcal{J}} \mathbb{E}_x \|f(x_{J^c}) - g(x_J)\|_2^2. \qquad (7)$$

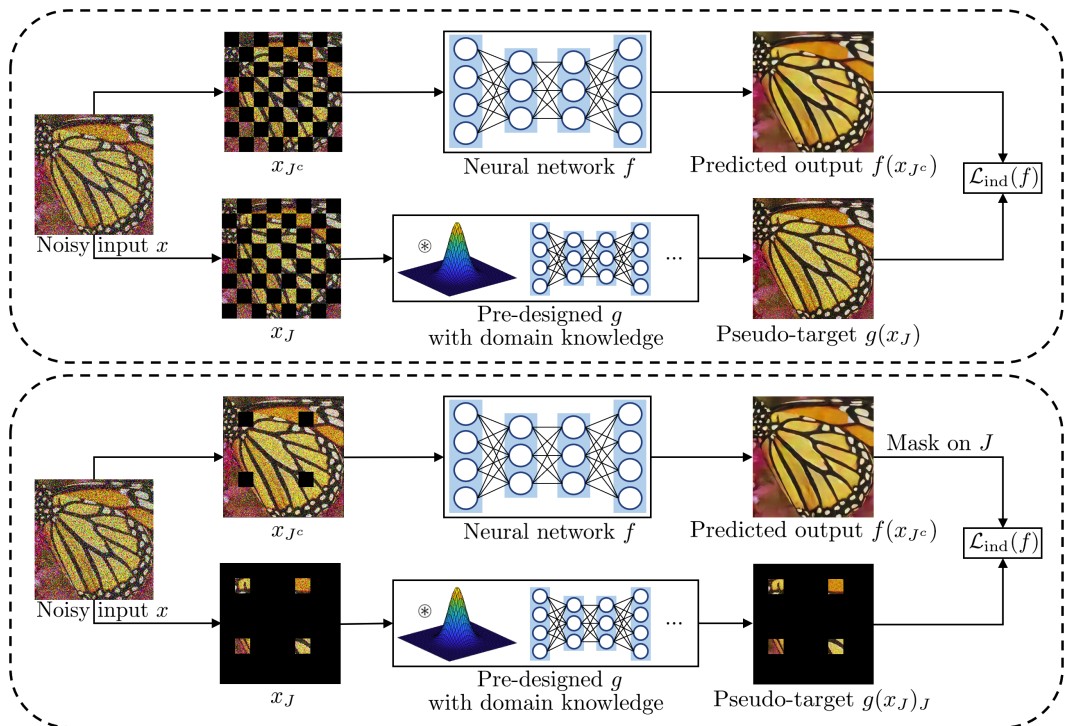

Figure 2: Proposed SSRL models using the $\mathcal{J}$-complement in denoiser learning. **Top:** $f$ and $g$ use almost equal amount of information from input, i.e., $|J| \approx |J^c|$, where $J$ and $J^c$ are complementary checkerboard masks. **Bottom:** $f$ and $g$ use unbalanced amount of information from input, specifically, $|J^c| \gg |J|$.

Figure 2 illustrates (7) with complementary checkerboard masks $J^c$ and $J$, where $f$ and $g$ use almost equal amount of information, and its variant, where $g$ uses much less information than $f$. The variant computes MSE only on $J \in \mathcal{J}$; in this setup, it is challenging for $g$ to predict the entire image.

**Relation to previous self-supervised denoising works.** In denoiser learning, the proposed SSRL loss (3) with the $\mathcal{J}$-complement of $f$ and $g = \mathcal{I}$ specializes to (2) with the $\mathcal{J}$-invariance of $f$, i.e., Noise2Self. Noise2Inverse (Hendriksen et al., 2020) and Neighbor2Neighbor (Huang et al., 2021) that emulate pairs of two independent noisy images by partitioning single measurements (e.g., CT ray measurements with independent noise and corrupted images with pixel-wise independent noise) can be viewed as Noise2Self. Thus, SSRL loss (3) with the $\mathcal{J}$-complement of $f$ and $g = \mathcal{I}$ specializes to the aforementioned Noise2Noise-motivated self-supervised denoising methods. (Noise2Noise also can be viewed by SSRL (3) with the setup above, by constructing $x$ with stacking two independent noisy images, where an image is corrupted by two independent noise realizations.)

### 2.4 SSRL USING DOMAIN KNOWLEDGE WITHOUT $\mathcal{J}$-COMPLEMENT

This section studies the SSRL loss (3) without using the $\mathcal{J}$-complement in Definition 1. Observe by (4) that $\mathbb{E}_{x,y}\|f(x) - y\|_2^2 + \|g(x_J) - y\|_2^2 = \mathbb{E}_x\|f(x) - g(x_J)\|_2^2 + 2\mathbb{E}_{x,y}\langle f(x) - y, g(x_J) - y\rangle$. Inspired by Noise2Same (Xie et al., 2020), the second proposed SSRL approach is to minimize an approximation of the right hand side in this equation that does *not* assume that $f$ and $g$ are $\mathcal{J}$-complementary. Our second main result finds an upper bound for the term $\mathbb{E}_{x,y}\langle f(x) - y, g(x_J) - y\rangle$ without relying on the $\mathcal{J}$-complement (remind that this term vanishes if $f$ and $g$ are $\mathcal{J}$-complementary; see Theorem 2).

**Theorem 3.** *Assume that* $\mathrm{Var}(g(x_J)_m|y) \leq \sigma^2, \forall m$. *Under Assumptions 1–3, the following bound holds:*

$$\mathbb{E}_{x,y}\langle f(x) - y, g(x_J) - y\rangle \leq \sigma\sqrt{M} \cdot \left(\mathbb{E}_x\|f(x) - f(x_{J^c})\|_2^2\right)^{1/2}. \tag{8}$$

*The following bound similarly holds for any $K \in \mathcal{K}$: $\mathbb{E}_{x,y}\langle f(x)_K - y_K, g(x_J)_K - y_K\rangle \le \sigma\sqrt{|K|} \cdot \left(\mathbb{E}_x\|f(x)_K - f(x_{J^c})_K)\|_2^2\right)^{1/2}$, where $K$ and $\mathcal{K}$ are defined in Theorem 2.*

*Proof.* See Section A.2 in the appendix. □

Using Theorem 3, the proposed SSRL loss that does not rely on the $\mathcal{J}$-complement is given by

$$\mathcal{L}(f) \triangleq \sum_{J \in \mathcal{J}} \mathbb{E}_x\|f(x) - g(x_J)\|_2^2 + 2\sigma\sqrt{M} \cdot \left(\mathbb{E}_x\|f(x) - f(x_{J^c})\|_2^2\right)^{1/2}, \tag{9}$$

where $\sigma$ is given in Theorem 3. Here, a regression NN $f$ can use information from the entire input $x$, whereas $\mathcal{L}_{\text{ind}}(f)$ in (7) uses only partial input $x_{J^c}$ in $f$. The intuition for designing $g$ in Section 2.3 similarly applies here. Our aim is to design good $g$ such that $g(x_J)$ is closer to $y$, consequently leading (9) close to (1). Similar to the variant of $\mathcal{L}_{\text{ind}}(f)$ (see Section 2.3), if the amount of information between two partitions $J$ and $J^c$ is unbalanced in denoiser learning, one can modify (9) to compute the MSE only on $J$ or $J^c$ in either both terms or the right term in (9). (These variants use the second result in Theorem 3.)

We conjecture that how well designed $g$ is, i.e., how close is pseudo-target $g(x_J)$ to ground-truth $y$, is captured by $\sigma$ defined in Theorem 3. We support the conjecture with examples in Section A.3 of the appendix. Then, this "goodness" of $g$ balances the two terms in (9) via $\sigma$. If $g$ is well-designed such that $g(x_J)$ is close to $y$, i.e., $\sigma^2$ is small, then the SSRL loss (9) relies more on the first term with good pseudo-target. If $g$ is poorly-designed such that $\sigma^2$ is large, then (9) puts more weight more on the second term that can implicitly promote the $\mathcal{J}$-invariance of $f$.

**Relation to previous self-supervised denoising work.** The proposed SSRL loss (9) becomes the Noise2Same loss (Xie et al., 2020, Thm. 2), by replacing $g(x_J)$ with $x$ and adding randomness to $K = J$ in the second term. In practice, one can tune $\sigma$ in (9) without knowing its exact value, similar to Noise2Same. One might use the aforementioned conjectured behavior of (9) with expected performance of $g$.

## 2.5 RELAXING NOISE ASSUMPTIONS OF EXISTING SELF-SUPERVISED DENOISING METHODS

This section explains how proposed SSRL (3) can relax noise assumptions of existing self-supervised denoising methods. Noise assumptions of existing self-supervised denoising methods include additive white Gaussian noise (AWGN with known variance) (Soltanayev & Chun, 2018), pixel-wise independent noise (Krull et al., 2019; Xie et al., 2020; Huang et al., 2021), and zero-mean noise (Lehtinen et al., 2018; Quan et al., 2020) or more generally $\mathbb{E}[x|y] = y$ (Batson & Royer, 2019; Xie et al., 2020). Assumption 1 of proposed SSRL relaxes the AWGN and pixel-wise independent noise assumptions, and is identical to the first assumption of Noise2Self (Batson & Royer, 2019). Assumption 2 of proposed SSRL can relax the second assumption of Noise2Self, $\mathbb{E}[x|y] = y$, that is also (implicitly) used in Noise2Same (Xie et al., 2020) and Noise2Noise (Lehtinen et al., 2018), by using a desinable function $g$. For example, $x$ is corrupted by additive non-zero-mean noise $e$ that is independent of $y$, i.e., $x = y + e$, then one can design $g$ as follows: $g(x) = x - \mathbb{E}[e]$, where $\mathbb{E}[e]$ can be estimated from calibration of imaging systems. The next section will explain how one can design $g$ using domain knowledge and select $g$ if domain knowledge is unavailable in image denoising.

## 3 EXAMPLES OF HOW TO DESIGN $g$ USING DOMAIN KNOWLEDGE, AND EMPIRICAL-LOSS APPROACH FOR SELECTING $g$ IN IMAGE DENOISING

Understanding noise statistics or properties is the first step towards accurate image recovery in computational imaging. First, this section describes how to use domain knowledge for designing $g$ in two imaging applications with practical noise models: *1) camera* image denoising in mixed Poisson–Gaussian–Bernoulli noise, *2)* low-dose CT denoising. Both applications have complicated noise models or strong noise, where Assumptions 1–2 in Section 2.2 are not completely satisfied. Noisy images in the first application are corrupted by independent and identically distributed (i.i.d.) noise with non-zero mean. The noise in the second application is approximately zero-mean but it is likely non-i.i.d. In designing $g$ for each application, we will use the suggestions in Sections 2.2–2.3, investigating Assumptions 1–3. Second, we propose a $g$-selection approach in image denoising that calculates some empirical measure related to (6) using only input training data.

### 3.1 DESIGNING $g$ USING DOMAIN KNOWLEDGE: CAMERA IMAGE DENOISING

The major noise sources in camera imaging (using charge coupled device) include object-dependent photoelectrons in image sensors, readout in camera electronics, and analog-to-digital converter and transmission errors that can be modeled by Poisson noise, AWGN, and Bernoulli (i.e., salt-and-pepper) noise models (Snyder et al., 1993), (Bovik, 2010, p. 90). We use the following practical mixed Poisson–Gaussian–Bernoulli noise model (Snyder et al., 1993; Batson & Royer, 2019):

$$x_n = \Pi_{[0,255]}(\text{Bernoulli}_p(\text{Poisson}(\lambda y_n)/\lambda + \epsilon_n)), \quad \epsilon \sim \mathcal{N}(0, \sigma_\epsilon^2 I), \quad n = 1, \dots, N, \quad (10)$$

where $\Pi_{[0,255]}$ performs 8-bit quantization and clips pixel values outside of $[0, 255]$, Bernoulli substitutes a pixel value with either 0 or 255 with probability $p$ (0 and 255 are coined with equal probability), Poisson generates pixel intensity-dependent Poisson noise with gain parameter $\lambda$, and $\epsilon$ is AWGN. Figure 3 (left) shows a noisy image corrupted by the mixed noise model (10). If an image $y$ is corrupted only by the mixed Poisson–Gaussian noise, $\mathbb{E}[x|y] = y$ in Assumption 2 can be satisfied ($\mathbb{E}[\text{Poisson}(\lambda y_n)/\lambda + \epsilon|y] = y_n, \forall n$). However, if Bernoulli noise is additionally considered as given in (10), the assumption $\mathbb{E}[x|y] = y$ will not be satisfied ($\mathbb{E}[\text{Bernoulli}_p(y_n)|y] = (1-p)y_n + 127.5p, \forall n$). The quantization-clipping operator $\Pi_{[0,255]}$ also makes it hard to satisfy $\mathbb{E}[x|y] = y$.

Following the suggestion based on Assumption 2 (see Section 2.2), we handcraft $g$ to "approximately" satisfy $\mathbb{E}[g(x)|y] = y$ with a simple operator. We interpret aforementioned Bernoulli noise and clipping artifact as salt-and-pepper noise. Median filtering is a computational efficient method that is effective in reducing salt-and-pepper and impulse noises (Bovik, 2010, §3.2). We design $g$ by applying weighted median filtering (Brownrigg, 1984) to a pixel with intensity either 0 or 255 at each color channel, aiming that this $g$ design "approximately" satisfy $\mathbb{E}[g(x)|y] = y$ by suppressing the salt-and-pepper noise effects cased by $\Pi_{[0,255]}$ and Bernoulli$_p$. (This is supported by empirical results in Section S.4.)

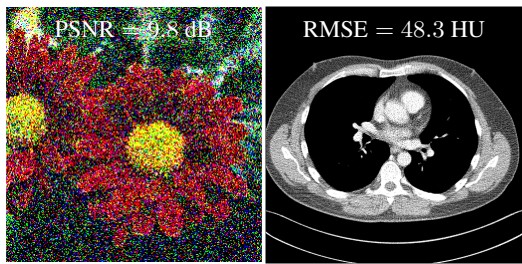

Figure 3: An input noisy image in camera image denoising in mixed noise **(left)** and low-dose CT **(right)**. PSNR and root mean square error (RMSE) values were averaged across all test samples.

Assumption 1 is satisfied because the noise in (10) is i.i.d. and independent of $y$. In Assumption 3, we conjecture that the above $g$ design is measurable (median operator is measurable under some conditions (Rustad, 2004)).

### 3.2 DESIGNING $g$ USING DOMAIN KNOWLEDGE: LOW-DOSE CT DENOISING

In X-ray CT (with a monoenergetic source), the pre-log measurement data is usually modeled by the Poisson model, i.e., $\text{Poisson}\{\rho_0 \exp(-[Ay]_l)\}, l = 1, \dots, L$, where $\rho_0$ is the number of incident photons per ray, $A \in \mathbb{R}^{L \times N}$ is a CT projection system matrix, and $L$ is the number of measured rays. Using the quadratic approximation to the log-likelihood of a Poisson model, the post-log measurement $z \in \mathbb{R}^L$ given $y$ can be approximated as the following Gaussian model (Sauer & Bouman, 1993; Fessler, 2000): $z|y \sim \mathcal{N}(Ay, C)$, where $C \in \mathbb{R}^{L \times L}$ is a *diagonal* covariance matrix and its diagonal elements become more nonuniform in lower-dose CT. This model suggests that post-log measurement may be modeled by $z = Ay + \varepsilon$, where $\varepsilon \sim \mathcal{N}(0, C)$. The filtered back-projection (FBP) method (Kak & Slaney, 1988, §3) performs computationally efficient CT reconstruction and has been widely used in commercial CT scanners (Pan et al., 2009). In low-dose CT, however, reconstructed image $x = Fz$ suffers from strong noise and streak artifacts, where $F \in \mathbb{R}^{N \times L}$ denotes a linear FBP operator, motivating research on learning denoising NNs. Figure 3 (right) shows a noisy FBP image in low-dose CT. Using the statistical results above, we model that a reconstructed image by $F$ is corrupted by an arbitrary additive noise $e$:

$$x = y + e, \quad e = (FA - I)y + F\varepsilon. \quad (11)$$

Low-dose CT uses all projection rays similar to standard-dose CT (but with substantially reduced dose) where $F$ approximately inverts $A$, i.e., $FA \approx I$, so we conclude that under (11), $\mathbb{E}[e] \approx 0$ and $\mathbb{E}[x|y] \approx y$. (See empirical results in Section S.4 that support $\mathbb{E}[e] \approx 0$.)

The above domain knowledge in low-dose CT indicates that handcrafting $g$ to have zero-mean $e$ to satisfy Assumption 2 can be redundant. Following the suggestion motivated by Theorem 2 (see Section 2.3), we set $g$ as a pre-trained denoiser by the existing self-supervised denoising methods (Batson & Royer, 2019; Hendriksen et al., 2020; Xie et al., 2020). Since such pre-trained $g$ will have some denoising capability and give better reference than $g = \mathcal{I}$ to (7) and (9) (see empirical results in Section S.4), we expect that proposed SSRL losses (7) and (9) improve the denoising quality over the aforementioned existing self-supervised denoising methods.

Assumption 1 is unlikely satisfied because in FBP images, neighboring noise components are likely to be correlated, i.e., $\text{Var}(e) \approx FCF^\top$ using $FA \approx I$ and noise model (11). Assumption 3 is satisfied as we use the conventional denoisiong NN, DnCNN (Zhang et al., 2017) and (modified) U-Net (Ronneberger et al., 2015), that are a continuous function.

### 3.3 EMPIRICAL-LOSS APPROACH FOR SELECTING $g$ IF DOMAIN KNOWLEDGE UNAVAILABLE

If accurate domain knowledge of a specific application is unavailable, it would be challenging to explicitly design $g$. In such cases in denoising, our general suggestion is to measure an existing self-supervised denoising loss, an upper bound of $\|g(x_J) - y\|_2^2$ or its variant that measure the quality of $g$, only with input training data. The lower quantity implies that $g$ is better and implicitly encapsulates better domain knowledge. In camera image denoising with the real-world dataset (Abdelhamed et al., 2018), the empirical measure of the Neighbor2Neighbor loss (Huang et al., 2021) $- \mathbb{E}\|g(x_J) - x_{J^c}\|_2^2$ – with setting $g$ as $\mathcal{I}$ and median filtering are $0.0052$ and $0.0048$, respectively. In low-dose CT denoising with the real-world dataset (Moen et al., 2021), the empirical measure of the Noise2Self loss (Batson & Royer, 2019) $- \mathbb{E}\|g(x_J)_{J^c} - x_{J^c}\|_2^2$ – with setting $g$ as $\mathcal{I}$ and pre-trained DnCNN by Noise2Self are $22044.5$ and $17062.0$ (in $\text{HU}^2$ where HU stands for modified Hounsfield unit), respectively. We expect better SSRL performance with the selected $g$ designs over $\mathcal{I}$.

## 4 EXPERIMENTAL RESULTS AND DISCUSSION

We evaluated proposed SSRL in two practical imaging applications in Section 3 with both simulated and real-world datasets. For these applications, we mainly focuses on comparisons with self-supervised denoising methods using single noisy input samples, particularly when statistical noise parameters are unavailable. We compared the performances of the following methods: Noise2Self (Batson & Royer, 2019), Noise2Noise-motivated methods that emulate pairs of two independent noisy images – Neighbor2Neighbor (Huang et al., 2021) or Noise2Inverse (Hendriksen et al., 2020) – Noise2Same (Xie et al., 2020), and corresponding SSRL to each aforementioned method. We also included Noise2True (1) results as baseline. For $f$ or $g$ in all the methods, we used the conventional denoising NN architecture, DnCNN (Zhang et al., 2017) or modified U-Net used in Noise2Self. We include experiment setup, and image and numerical results for/from real-world datasets in Sec. A.4.

### 4.1 EXPERIMENTAL SETUP FOR SIMULATED DATASETS

**Camera image denoising in mixed noise.** We evaluated the proposed SSRL framework with three RGB camera image datasets, ImageNet ILSVRC 2012 Val (Russakovsky et al., 2015), BSD 300 (Martin et al., 2001), Set 5 (Bevilacqua et al., 2012). For training, we used the ImageNet ILSVRC 2012 Val dataset with $20,000$ images; for tests, we used the BSD 300 and Set 5 datasets consisting of $300$ and $5$ images, respectively. We simulated noisy images with the following imaging parameters introduced in (10): $\lambda = 30$, $\sigma_\epsilon = 60$, and $p = 0.2$. We evaluated the denoising quality by the most conventional error metric in camera image denoising, PSNR and structural similarity index measure (SSIM). In Neighbor2Neighbor setup, we emulated two independent noisy images from single noisy images by random neighbor sub-sampling with $2 \times 2$-window (Huang et al., 2021).

**Low-dose CT denoising.** We evaluated the proposed SSRL framework with The 2016 Low Dose CT Grand Challenge data (McCollough, 2016). We selected $200$ regular-dose chest images of size $N = 512 \times 512$ and the 3 mm slice thickness from four patients. For training, we used $170$ ($85\%$) chest images from three patients; for tests, we used $30$ ($15\%$) chest images from the other patient. We simulated low-dose sinograms using the Poisson model with the selected regular-dose chest datasets. In particular, we simulated sinograms of size $L = 736 \times 1152$ ('detectors' $\times$ 'projection views'),

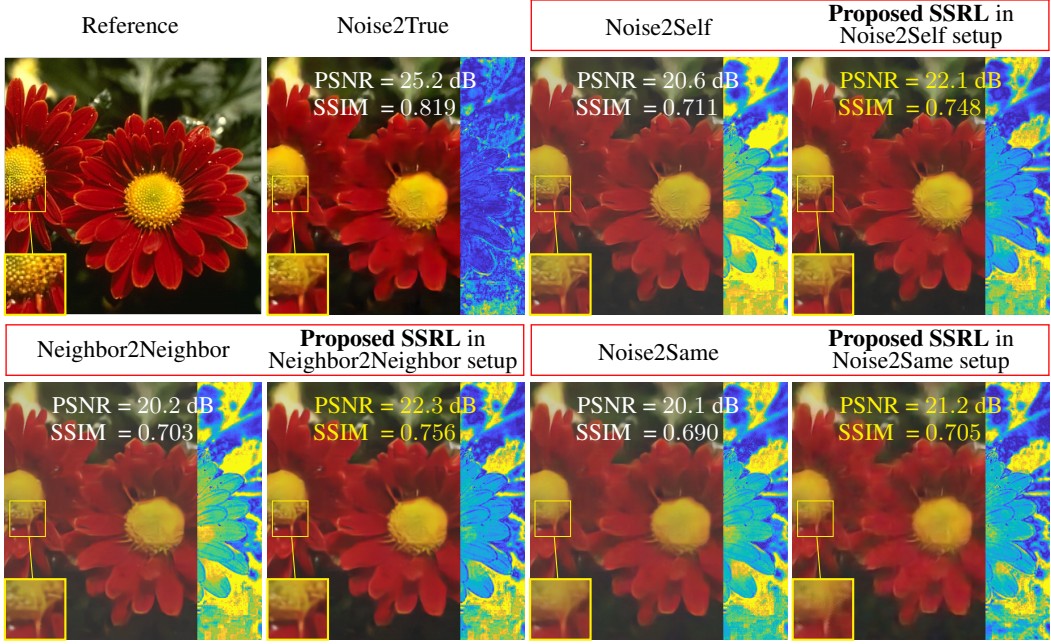

Figure 4: Comparisons of denoised images **(left)** via DnCNNs from different learning methods and their saturation error maps **(right)** in camera image denoising (blue and yellow denote $0$ and $0.5$ absolute error, respectively). PSNR & SSIM values were averaged across all BSD 300 test samples.

with fan-beam geometry corresponding to a no-scatter monoenergetic source with $\rho_0 = 5 \times 10^4$. We used FBP (Kak & Slaney, 1988, §3) to reconstruct images with resolution $0.69$ mm $\times$ $0.69$ mm. We evaluated the denoising quality by the most conventional error metric in CT application, RMSE in HU. In Noise2Inverse setup, we emulated two independent noisy images by partitioning single sinograms with odd and even views and applying FBP to two partitioned independent sinograms.

## 4.2 COMPARISONS BETWEEN DIFFERENT SELF-SUPERVISED DENOISING METHODS

Compare each existing self-supervised denoising method to its corresponding SSRL setup in Figures 4–5, and Figures S.1–S.4 and Tables S.1–S.3; see three comparison sets, each grouped by red box. For both applications, proposed SSRL achieves significantly better image denoising quality, i.e., closer to the Noise2True quality, compared to the existing methods, Noise2Self, Neighbor2Neighbor, Noise2Inverse, and Noise2Same, regardless of the regression NN architecture. We show DnCNN prediction uncertainty of all the methods in Figure S.5.

**Camera image denoising in mixed noise (simulated data).** Figures 4 and S.3 show that in all the three comparison sets, SSRL gives closer image quality, particularly color saturation, to Noise2True than existing methods, Noise2Self, Neighbor2Neighbor, and Noise2Same. Setting $g$ as weighted median filtering avoids bias in SSRL loss caused by salt-and-pepper noise. Yet, compared to Noise2True, denoised images obtained by proposed SSRL lack saturation and detail preservation. For saturation and detail preservation comparisons, see Figures 4, S.1 and S.3.

Comparing the three comparison sets in Figures 4 and S.3, and Tables S.1–S.2 shows that all the Noise2Self, Neighbor2Neighbor, and Noise2Self setups have comparable results in terms of PSNR values. The potential reason is that in this application, all the three setups similarly satisfy Assumptions 1–3 (see Section 2.2); in particular, Assumption 1 is well-satisfied by pixel-wise i.i.d. noise.

In this application, the zero-mean noise assumption of the existing self-supervised denoising methods is violated, whereas its counterpart in proposed SSRL, Assumption 2, is "approximately" satisfied by $g$ in Section 3.1. This suggests the importance of satisfying Assumption 2 with good $g$.

**Low-dose CT denoising (simulated data).** In all the three comparison sets, SSRL better recovers low-contrast regions (e.g., soft tissues) and small details, and significantly reduces noise and artifacts throughout the image, over existing methods, Noise2Self, Noise2Inverse, and Noise2Same. See

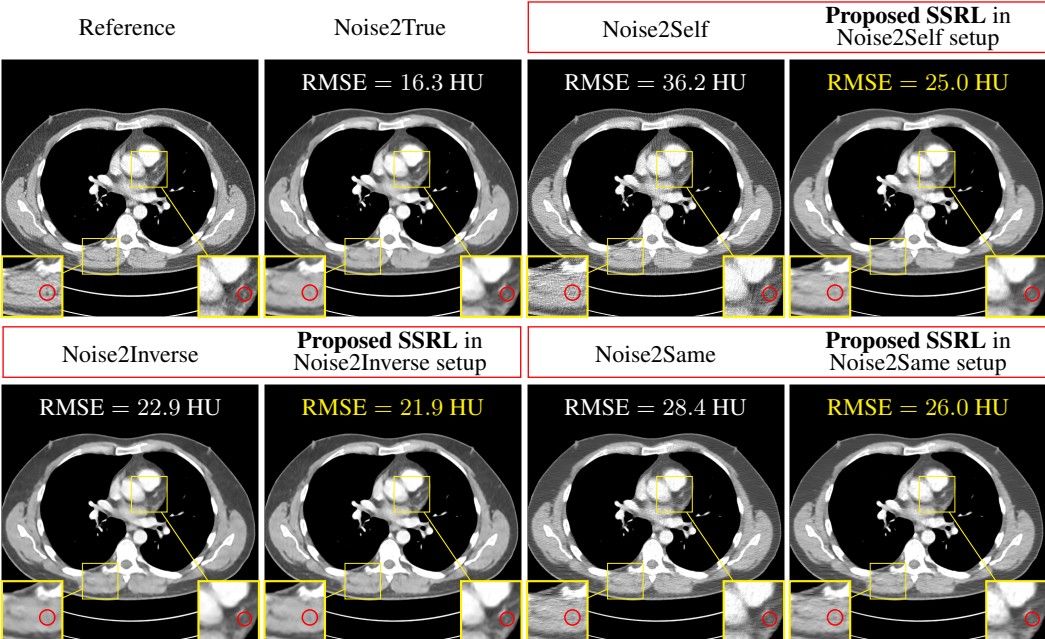

Figure 5: Comparisons of denoised images via DnCNNs from different learning methods in low-dose CT (display window is $[800, 1200]$ HU). RMSE values were averaged across all test samples.

zoom-ins and circled small details in Figures 5 and S.4, and error images in Figure S.2, particularly in 'Noise2Self vs. Propose SSRL in Noise2Self setup' and 'Noise2Same vs. Proposed SSRL in Noise2Same setup.' The results might imply that simply setting $g$ as pre-trained NN by existing self-supervised denoising methods works like a charm in SSRL. Proposed SSRL in the Noise2Inverse setup can provide images with image quality that is comparable to conventional FBP at 10 times higher dose (when $\rho_0 = 5 \times 10^5$, RMSE = 20.5 HU on average; see DnCNN results in Figure 5).

Next, comparing Noise2Self and Noise2Same result sets to that of Noise2Inverse in Figures 5, S.2 and S.4, and Table S.3 shows that Noise2Inverse setup significantly improves the denoising quality, compared to Noise2Self and Noise2Same setups. We conjecture that violation of Assumption 1 – that is satisfied in the Noise2Inverse setup but unlikely to be satisfied in the Noise2Self and Noise2Same setups in this application – degrades the performance.

**Camera image denoising and low-dose CT denoising with real-world datasets.** Denoised image results in Figures A.2–A.3 and S.8–S.9 from the two real-world datasets (Abdelhamed et al., 2018; Moen et al., 2021) demonstrate that SSRL improves existing self-supervised denoising methods, particularly Neighbor2Neighbor, Noise2Self, and Noise2Same, *without* having their exact noise properties. The results well correspond to our expectation in Section 3.3.

## 5 CONCLUSION

It is important to develop SSRL that enables comparable prediction performances to supervised learning, because it is extremely challenging to collect many ground-truth target samples in many practical computational imaging and computer vision applications. The proposed SSRL framework bridges the gap between SSRL and supervised regression learning via domain knowledge of applications. To achieve closer prediction performance to supervised learning, SSRL uses domain knowledge to design a better pseudo-predictor $g$ such that $g(x_J)$ becomes closer to $y$. For camera image denoising and low-dose CT denoising with both simulated and real-world datasets, SSRL achieves more accurate prediction compared to the existing self-supervised denoising methods (Batson & Royer, 2019; Huang et al., 2021; Hendriksen et al., 2020; Xie et al., 2020). Remark, however, that applying SSRL to other regressions problems may need careful investigations about $g$ based on their domain knowledge. Our future work is extending proposed SSRL to other machine learning problems such as teacher-student models (see Section S.7) and meta-learning. On the application side, our future work is applying SSRL to regression problems beyond image denoising.

## 6    REPRODUCIBILITY

Section 2.2 specifies all the theoretical assumptions. Section A.1 and Section A.2 in the appendix include the complete proofs of Theorem 2 and Theorem 3, respectively. Section S.2 in the supplement includes the complete implementation details including the hyperparameter selection strategies and the chosen hyperparameters. We included an anonymized zip file that includes test codes, test data, trained models, and instructions and codes that provide complete description of the data processing steps, as supplementary materials. We will make our codes (for data construction, training, and test) and trained models publicly available on GitHub if the paper is accepted.

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

APPENDIX

## A.1 PROOFS FOR THEOREM 2

Observe first that combining Assumptions 1 & 3 and the $\mathcal{J}$-complement implies that $f(x)_m$ and $g(x)_m$ are conditionally independent, given $y$, i.e.,

$$f(x_{J^c})_m | y \perp\!\!\!\perp g(x_J)_m | y, \quad \forall m.$$

Using this result with Assumption 2 and reminding that the $\mathcal{J}$-complement implies $\mathbb{E}_x \| f(x) - g(x) \|_2^2 = \mathbb{E}_x \| f(x_{J^c}) - g(x_J) \|_2^2$, we obtain the following result from (4):

$$\mathbb{E}_x \| f(x_{J^c}) - g(x_J) \|_2^2 = \mathbb{E}_{x,y} \| f(x_{J^c}) - y \|_2^2 + \| g(x_J) - y \|_2^2$$

where the equality uses

$$\begin{aligned}
\mathbb{E}_{x,y} \langle f(x_{J^c}) - y, g(x_J) - y \rangle &= \mathbb{E}_y \mathbb{E}_{x|y} \langle f(x_{J^c}) - y, g(x_J) - y \rangle \\
&= \sum_m \mathbb{E}_y (\mathbb{E}_{x|y}[f(x_{J^c})_m - y_m])(\mathbb{E}_{x|y}[g(x_J)_m - y_m]) \\
&= 0
\end{aligned}$$

in which the second equality uses the first result above and the third equality holds by Assumption 2. This completes the proofs.

## A.2 PROOFS FOR THEOREM 3

We first obtain the following bound:

$$\begin{aligned}
&\mathbb{E}_{x,y} \langle f(x) - y, g(x_J) - y \rangle \\
&= \mathbb{E}_y \mathbb{E}_{x|y} \sum_m (f(x)_m - y_m)(g(x_J)_m - y_m) \\
&= \sum_m \mathbb{E}_y \left[ \mathbb{E}_{x|y}(f(x)_m - y_m)(g(x_J)_m - y_m) - \mathbb{E}_{x|y}(f(x)_m - y_m)\mathbb{E}_{x|y}(g(x_J)_m - y_m) \right] \\
&= \sum_m \mathbb{E}_y \left[ \mathrm{Cov}(f(x)_m - y_m, g(x_J)_m - y_m | y) \right] \\
&= \sum_m \mathbb{E}_y \left[ \mathrm{Cov}(f(x)_m, g(x_J)_m | y) \right] \\
&= \sum_m \mathbb{E}_y \left[ \mathrm{Cov}(f(x)_m - f(x_{J^c})_m, g(x_J)_m | y) \right] \\
&\leq \sum_m \mathbb{E}_y \left[ (\mathrm{Var}(f(x)_m - f(x_{J^c})_m | y) \cdot \mathrm{Var}(g(x_J)_m | y))^{1/2} \right] \\
&\leq \sum_m \mathbb{E}_y \left[ \mathrm{Var}(f(x)_m - f(x_{J^c})_m | y) \cdot \mathrm{Var}(g(x_J)_m | y) \right]^{1/2} \\
&\leq \left( M \cdot \sum_{m=1}^M \mathbb{E}_y \left[ \mathrm{Var}(f(x)_m - f(x_{J^c})_m | y) \cdot \mathrm{Var}(g(x_J)_m | y) \right] \right)^{1/2} \\
&\leq \left( M \cdot \sum_{m=1}^M \mathbb{E}_y \left[ \mathrm{Var}(f(x)_m - f(x_{J^c})_m | y) \cdot \sigma^2 \right] \right)^{1/2},
\end{aligned}$$

where the second equality holds by Assumption 2, the third equality uses $\mathrm{Cov}(X, Y | Z) = \mathbb{E}[XY|Z] - \mathbb{E}[X|Z]\mathbb{E}[Y|Z]$ where $X$, $Y$, and $Z$ are random variables or vectors, the fifth equality holds because $f(x_{J^c})_m$ does not correlate with $g(x_J)_m$, $\forall m$ (due to Assumptions 1 and 3), so subtracting $f(x_{J^c})_m$ from $f(x)_m$ does not change the covariance. Now, the first inequality uses the Pearson correlation coefficient bound, the second inequality uses the Jensen's inequality $\mathbb{E}\sqrt{X} \leq \sqrt{\mathbb{E}X}$, the third inequality uses the Jensen's inequality $\sum_m \sqrt{a_m} \leq \sqrt{M' \sum_m a_m}$ for

any $a \in \mathbb{R}^{M'}$, and the last inequality holds by the conditional variance bound specified in Theorem 3. We bound and rewrite the final result above and this completes the proof:

$$
\begin{aligned}
\mathbb{E}_{x,y} \langle f(x) - y, g(x_J) - y \rangle &\leq \sigma \sqrt{M} \cdot \left( \sum_{m=1}^{M} \mathbb{E}_y \left[ \mathrm{Var}(f(x)_m - f(x_{J^c})_m | y) \right] \right)^{1/2} \\
&\leq \sigma \sqrt{M} \cdot \left( \sum_{m=1}^{M} \mathbb{E}_y \left[ \mathbb{E}_{x|y} \left[ f(x)_m - f(x_{J^c})_m \right]^2 \right] \right)^{1/2} \\
&= \sigma \sqrt{M} \cdot \left( \sum_{m=1}^{M} \mathbb{E}_x \left[ f(x)_m - f(x_{J^c})_m \right]^2 \right)^{1/2},
\end{aligned}
$$

where the equality uses the filtration property of conditional expectation.

## A.3 EXAMPLES THAT SUPPORT CONJECTURED BEHAVIOR OF (9) WITH $\sigma$

The first example in general regression models the pseudo-target as follows: $g(x_J) = y + e_1$, where $e_1 \in \mathbb{R}^M$ is some arbitrarily additive noise independent of $y$. This gives $\mathrm{Var}(g(x_J)_m | y) = \mathrm{Var}(y_m + (e_1)_m | y) = \mathrm{Var}(e_1)_m \leq \sigma^2, m = 1, \ldots, M$. Under this model, how close is $g(x_J)$ to $y$ is captured by $\sigma$.

The second example in image denoising assumes that $x$ is corrupted by AWGN $e_2 \in \mathbb{R}^N$ that is independent of $y$. Setting $g$ as a linear mapping $G \in \mathbb{R}^{J \times N}$ gives $\mathrm{Var}(g(x_J)_n | y) = \mathrm{Var}((Gy_J)_n + (G(e_2)_J)_n | y) = \mathrm{Var}((G(e_2)_J)_n) \leq \sigma^2, \forall n = 1, \ldots, N$. Under this model, how close is $g(x_J)$ to $y$ is captured by $\sigma$.

## A.4 EXPERIMENTAL SETUP AND RESULTS FOR/FROM REAL-WORLD DATASETS

### A.4.1 EXPERIMENTAL SETUP FOR REAL-WORLD DATASETS

We also evaluated the proposed SSRL framework with real-world camera image and low-dose CT datasets, where we do not have their complete noise properties/statistics. We chose the publicly available SIDD sRGB Data (Abdelhamed et al., 2018) and Low Dose CT Image and Projection Data (Moen et al., 2021), where both the datasets include high-quality images or standard-dose FBP images so that one can run Noise2True experiments and obtain quantitative results. We used the DnCNN architecture for all experiments. For each experiment, we used the same implementation setup (such as hyperparameters and masking scheme) as that in the corresponding simulated data experiment. In particular, $g$ is median filter and pre-trained denoiser via existing self-supervised denoising methods in camera image denoising and low-dose CT denoising experiments, respectively.

**Camera image denoising with SIDD sRGB Data.** We used the SIDD sRGB training and validation datasets for training and tests, respectively. We chose the representative comparison setup, Neighbor2Neighbor, from the camera image denoising experiments using simulated data in Section 4.1.

**Low-dose CT denoising with Low Dose CT Image and Projection Data.** We followed the data construction setup in Section 4.1 that was used in simulated data experiments; we remark that chest CT scans in the Low Dose CT Image and Projection Data use the 1 mm slice

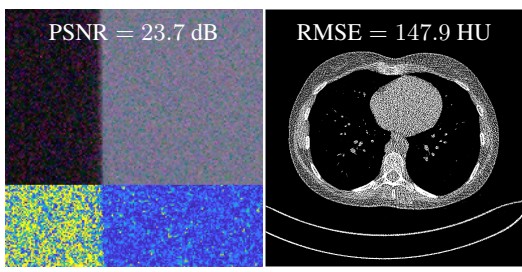

Figure A.1: An input intrinsically-noisy image in camera image denoising (**left**) and low-dose CT (**right**). PSNR and RMSE values were averaged across all test samples.

thickness. We cannot run Noise2Inverse experiments because the Low Dose CT Image and Projection Data does not provide two independent half-view FBP images. We thus chose the other comparison setups, Noise2Self and Noise2Same.

### A.4.2 MAIN EXPERIMENTAL RESULTS FROM REAL-WORLD DATASETS

This section includes main experimental results such as denoised images and calculated performance measure, from the real-world datasets. Section S.3.2 in the supplement includes their supplementary results.

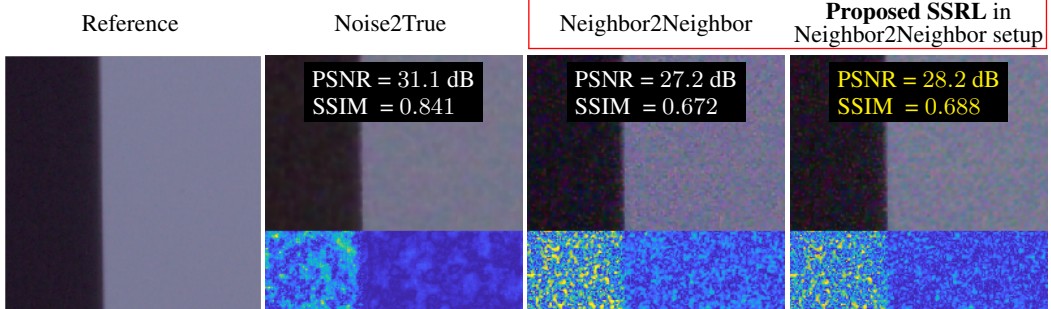

Figure A.2: Comparisons of denoised images **(top)** via DnCNNs from different learning methods and their saturation error maps **(bottom)** in camera image denoising (blue and yellow denote 0 and 0.5 absolute errors, respectively). PSNR and SSIM values were averaged across all SIDD sRGB validation samples.

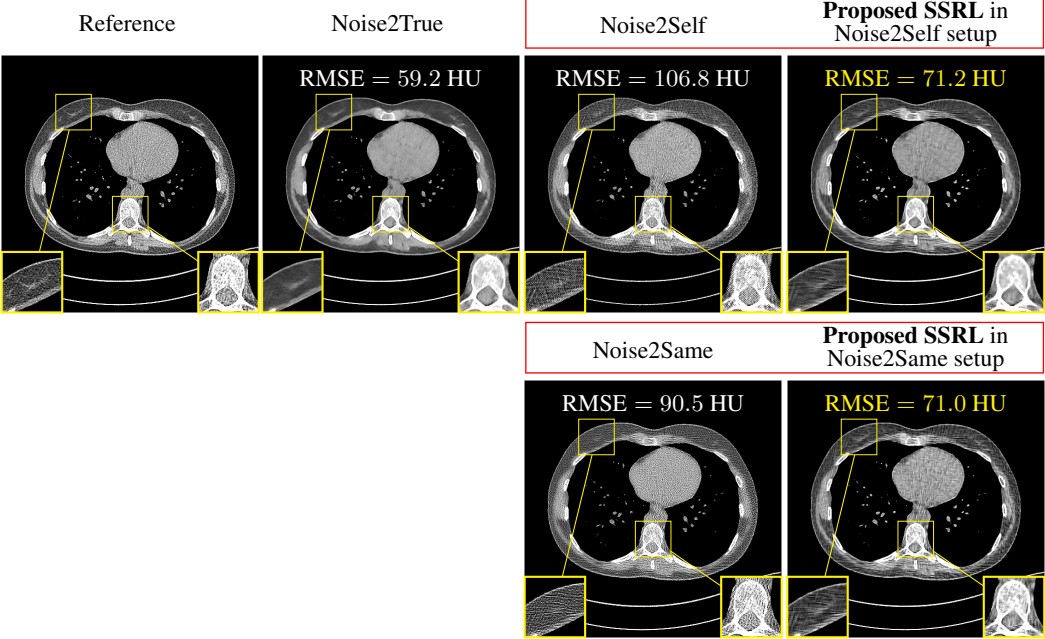

Figure A.3: Comparisons of denoised images via DnCNNs from different learning methods in low-dose CT (display window is $[800, 1200]$ HU). RMSE values were averaged across all test samples.

# Self-supervised regression learning using domain knowledge: Applications to improving self-supervised image denoising (Supplement)

**Anonymous authors**

## S.1 Detailed paper contributions

This section elaborates the contributions of the proposed SSRL framework:

1. The paper applies the proposed SSRL generalization to several recent representative self-supervised denoising methods, Noise2Self (Batson & Royer, 2019), Noise2Same (Xie et al., 2020), Noise2Noise (Lehtinen et al., 2018), Noise2Inverse (Hendriksen et al., 2020), and Neighbor2Neighbor (Huang et al., 2021). See Sections 2.3–2.4. With camera image and low-dose CT denoising experiments using both real and synthetic datasets, the paper demonstrates the outperforming performance of SSRL extensions over the aforementioned self-supervised denoising methods.

2. Section 2.5 shows that designable pseudo-predictor $g$ in SSRL can relax noise assumptions of existing self-supervised denoising methods. In addition, Section 4.2 includes experiments studying how denoising performances change with satisfying noise assumption(s) (i.e., Assumptions 1–2).

3. The paper explains how to incorporate domain knowledge into self-supervised denoising methods via $g$ and why more accurate domain knowledge can improve them. See examples in Sections 3.1–3.2. In addition, Section 3.3 proposes an empirical approach for selecting $g$ if domain knowledge of specific applications is unavailable.

4. The proposed SSRL framework in Section 2 considers regression NN learning beyond denoiser learning, by using a desinable operator $g$. The paper is the first step towards self-supervised learning in regression problems, by showing that the proposed framework extends well to image denoising problem.

## S.2 Implementation details, data and code licenses, and library versions

This section describes implementation details, lists hyperparameters, and specifies license of datasets and codes used in this study.

### S.2.1 Data and code licences, and library versions

The ImageNet ILSVRC 2012 Val and BSD 300 datasets have the Custom license (research, non-commercial), the Set 5 data has the Unknown license, and the SIDD sRGB Data (Abdelhamed et al., 2018) has the MIT license. We obtained The 2016 Low Dose CT Grand Challenge data (McCollough, 2016) from https://aapm.app.box.com/s/eaw4jddb53keg1bptavvvd1sf4x3pe9h/file/856956352254, and Low Dose CT Image and Projection Data (Moen et al., 2021) from https://doi.org/10.7937/9npb-2637. The 2016 Low Dose CT Grand Challenge data and Low Dose CT Image and Projection Data have the Custom license and the patient information is fully redacted.

We implemented all the methods specified in Section 4 by modifying the Noise2Self code (Batson & Royer, 2019) (GitHub repository: https://github.com/czbiohub/noise2self with

version Dec. 17, 2019) that is licensed under the MIT license. For all training and testing experiments, we used Pytorch 1.0.0 or 1.7.0 (Paszke et al., 2019) with the BSD-style license. For simulating low-dose FBP images, we used the Michigan image reconstruction toolbox (MIRT) (Fessler, 2016) of which license information is declared on its release page. For sinogram generation and FBP reconstruction, we used the "`Gtomo2_dscmex.m`" routine (updated on Dec. 10, 2006) and the "`fbp2.m`" routine (updated on Dec. 21, 2005), respectively.

### S.2.2 COMMON IMPLEMENTATION DETAILS IN BOTH APPLICATIONS

For all the existing self-supervised denoising methods specified in Section 4 and Noise2True, we finely tuned their hyperparameters, including the initial learning rate, learning rate decay parameters, minibatch size, number of DnCNN layers, and balancing parameter $\sigma$ (see, e.g., (9)), to achieve the best numerical results. We simply applied the chosen hyperparameter sets to corresponding SSRL setups. We applied the chosen learning rate decay parameters, minibatch size, and number of DnCNN layers in Noise2True experiments to all the self-supervised denoising methods.

For the existing self-supervised denoising methods, Noise2Self (Batson & Royer, 2019) and Noise2Same (Xie et al., 2020), we used their default masking setups. The Noise2Self default setup uses the deterministic masking scheme for each $J$ that equi-spacedly samples $6.25\%$ of the number of pixels in each training image (i.e., a single pixel is selected in each $4 \times 4$ window). The Noise2Same default setup uses the saturated sampling scheme (Xie et al., 2020; Krull et al., 2019) for each $J$ that randomly samples $\approx 0.5\%$ of the number of pixels in each training image (i.e., a single pixel is sampled in each $14 \times 14$ window). In training denoising NNs, both methods interpolate missing pixels in $x_{J^c}$ by applying weighted average to their 8 neighboring pixels, and use interpolated $x_{J^c}$ as input to denoisers.

For the existing Noise2Noise-motivated methods that emulate pairs of two independent noisy images, Neighbor2Neighbor (Huang et al., 2021) and Noise2Inverse (Hendriksen et al., 2020), we calculated their loss with non-masked images as proposed.

We tested all trained regression NNs to non-masked images – rather than masked images with $J^c$ – as this setup gave higher denoising accuracy than prediction with masked images (Batson & Royer, 2019; Xie et al., 2020).

### S.2.3 IMPLEMENTATION DETAILS FOR EXPERIMENTS WITH SYNTHETIC CAMERA IMAGE DENOISING DATA

The common hyperparameters for all learning methods were defined as follows. (In this application, these gave good image denoising performance across all existing self-supervised denoising methods and Noise2True, since we rescaled or normalized training images; see details below.) We used the default 17-layer DnCNN (Zhang et al., 2017) and the modified U-Net used in Noise2Self (Batson & Royer, 2019), and trained all DnCNNs and U-Nets with the mini-batch version of Adam (Kingma & Ba, 2015). We selected the initial learning rate, the batch size, and the number of epochs as $8 \times 10^{-4}$, 8, and 190, respectively, and decayed the learning rates by a factor of $0.5$ every 50,000 iterations. (For Neighbor2Neighbor (Huang et al., 2021), we set the batch size as 32, as it reduces training image size with $2 \times 2$ sub-sampling window.) We used the data augmentations in Noise2Same (Xie et al., 2020), i.e., random crops with size $256 \times 256$, rotation and flipping. Except for Noise2Same, we rescaled all images to $[0, 1]$, following (Batson & Royer, 2019; Huang et al., 2021). In Noise2Same experiments (including SSRL-Noise2Same), we normalized each image by subtracting its mean and dividing by its standard deviation at each channel, following (Xie et al., 2020).

For proposed SSRL in the Noise2Self and Noise2Same setups (referred to as SSRL-Noise2Self and SSRL-Noise2Same, respectively), we used the deterministic masking scheme in Figure 2(bottom) for each $J$ with $\approx 11.1\%$ and $\approx 1.2\%$ sampling ratio, respectively – i.e., a single pixel is selected in each $3 \times 3$ and $9 \times 9$ window, respectively. These setups gave more appealing results than the default masking parameters (i.e., $4 \times 4$ window in Noise2Self and $14 \times 14$ window in Noise2Same); compare Figure S.6 to corresponding results in Figure 4. We observed in this application that using sufficient amount of information for a linear interpolation in $f$ is useful for giving good prediction. For weighted median filtering (Brownrigg, 1984) $g$ in all SSRL setups, we used the following weights:

$[1, 2, 1; 2, \boxed{9}, 2; 1, 2, 1]$, where a box denotes the central weight. The dilation rates of weighted median filtering for SSRL in the Noise2Self, Noise2Same, and Neighbor2Neighbor setups are 3, 9, and 1, respectively, corresponding to the distances between pixels in $J$. For SSRL-Noise2Self, we computed $\mathcal{L}_{\text{ind}}$ in (7) only on $J$ (see Figure 2(bottom)). For SSRL-Noise2Same, we used the same balancing parameter $\sigma$ as Noise2Same used, i.e., $\sigma = 1$, and computed $\mathcal{L}$ (both terms) in (9) only on $J$.

The DnCNN and U-Net training time for each experiment was less than 72 hours with an NVIDIA TITAN V GPU.

### S.2.4 IMPLEMENTATION DETAILS FOR EXPERIMENTS WITH SYNTHETIC LOW-DOSE CT DENOISING DATA

We used fan-beam geometry for sinogram simulation, where width of each detector column is 1.2858 mm, source to detector distance is 1085.6 mm, and source to rotation center distance is 595 mm. For the FBP method, we used a ramp filter because in general, it better preserves the sharpness of edges on reconstructed images than Hanning filter (but overall noise increases).

The common hyperparameters for all learning methods were defined as follows. We used 8-layer DnCNN (Zhang et al., 2017) with its default setup and the modified U-Net used in Noise2Self (Batson & Royer, 2019), and trained all DnCNNs and U-Nets with the mini-batch version of Adam (Kingma & Ba, 2015). We selected the batch size and the number of epochs as 2 and 1,000, respectively, and decayed the learning rates by a factor of 0.95 every 10 epochs. In training DnCNNs and U-Nets, we selected the initial learning rates as 0.1 and $5 \times 10^{-5}$, respectively, unless stated otherwise.

For proposed SSRL-Noise2Self and SSRL-Noise2Same, we used complementary checkboard masks $J$ and $J^c$ in Figure 2(top). We observed in this application that if $g$ is set to use small amount of information, i.e., $|J| \ll |J^c|$, then pre-trained $g$ makes poor prediction. For SSRL-Noise2Self, we set $g$ as pre-trained NN by Noise2Self with complementary checkerboard masks (see its inference results with 8-layer DnCNN and U-Net in Figures S.7(a) and S.4, respectively). In training DnCNNs, we used the same initial learning rate as that used in pre-training $g$, 0.01. We computed $\mathcal{L}_{\text{ind}}$ as given in (7) (see Figure 2(top)).

For SSRL in the Noise2Inverse setup, we set $f$ and $g$ as $f/2$ and $\mathcal{I} - g/2$, respectively, where $g$ is pre-trained NN by Noise2Inverse. In training DnCNNs, we used the same initial learning rate as that used in pre-training $g$, 0.001. In inference, we averaged the predictions from $f$ and $g$, as this corresponds to training setup above. In all Noise2Inverse inferences (including SSRL-Noise2Inverse), we input full-view FBP images since this is consistent with other experiments and gave better denoising performance than denoising half-view FBP images. For SSRL-Noise2Same, we set $g$ as pre-trained NN by Noise2Same with complementary checkerboard masks (see its test results with DnCNN and U-Net in Figures S.7(b) and S.4, respectively), and computed $\mathcal{L}$ (both terms) in (9) only on $J^c$. We chose the balancing parameter $\sigma$ as 15, setting the ratio of the first term to the squared second term $2\sigma\sqrt{M}\|f(x)_{J^c} - f(x_{J^c})_{J^c}\|_2^2$ in (9) as 10. For Noise2Same with either the default setup and complementary checkerboard masks, we chose the balancing parameter $\sigma$ as 500. For self-supervised denoising methods with complementary checkerboard masks, we interpolated missing pixels in both $x_{J^c}$ and $x_J$, by averaging their 4 neighboring pixels.

The DnCNN and U-Net training time for existing self-supervised denoising methods and proposed SSRL methods was less than 10 hours and 12 hours, respectively, with an NVIDIA TITAN Xp GPU. It took total less than 22 hours to train both $f$ and $g$.

### S.3 SUPPLEMENTARY EXPERIMENTAL RESULTS FOR SECTION 4

This section mainly includes supplementary materials to Section 4.

#### S.3.1 SUPPLEMENTARY EXPERIMENTAL RESULTS WITH SYNTHETIC DATASETS

Figures S.1 and S.2 show error maps of denoised images via DnCNNs from different learning methods in camera image denoising in mixed noise and low-dose CT, respectively. These show for both

applications that in all the three comparison setups, SSRL significantly reduces errors and artifacts across the entire image, over existing self-supervised denoising methods. Red boxes compare existing self-supervised denoising method to proposed SSRL in the corresponding setup.

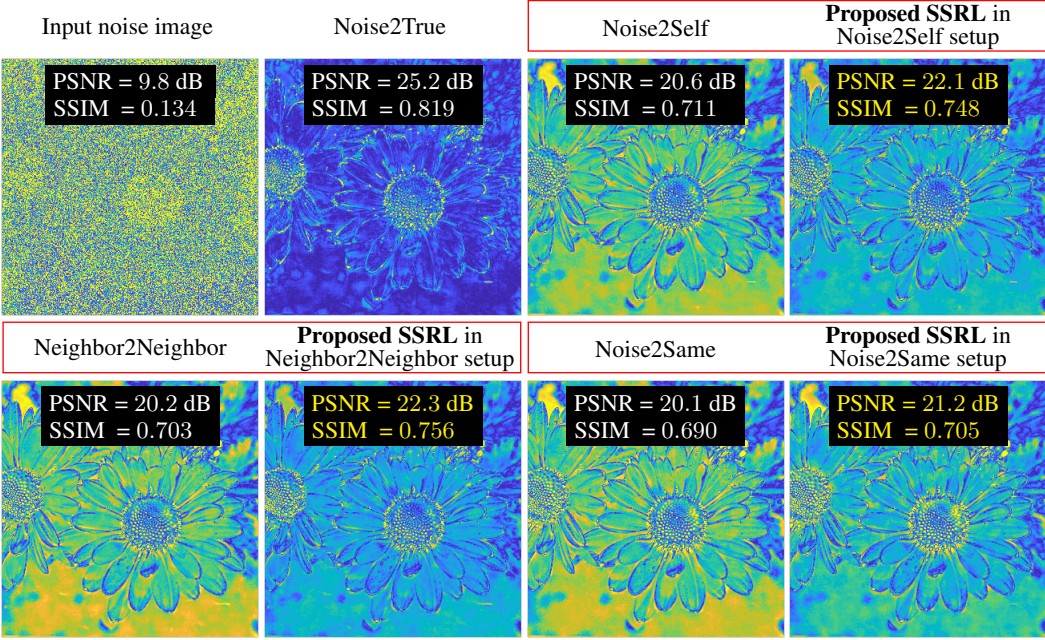

Figure S.1: Error map comparisons of denoised images via DnCNNs from different learning methods in camera image denoising (blue and yellow denote 0 and 50 absolute errors, respectively). PSNR and SSIM values were averaged across all BSD 300 test samples.

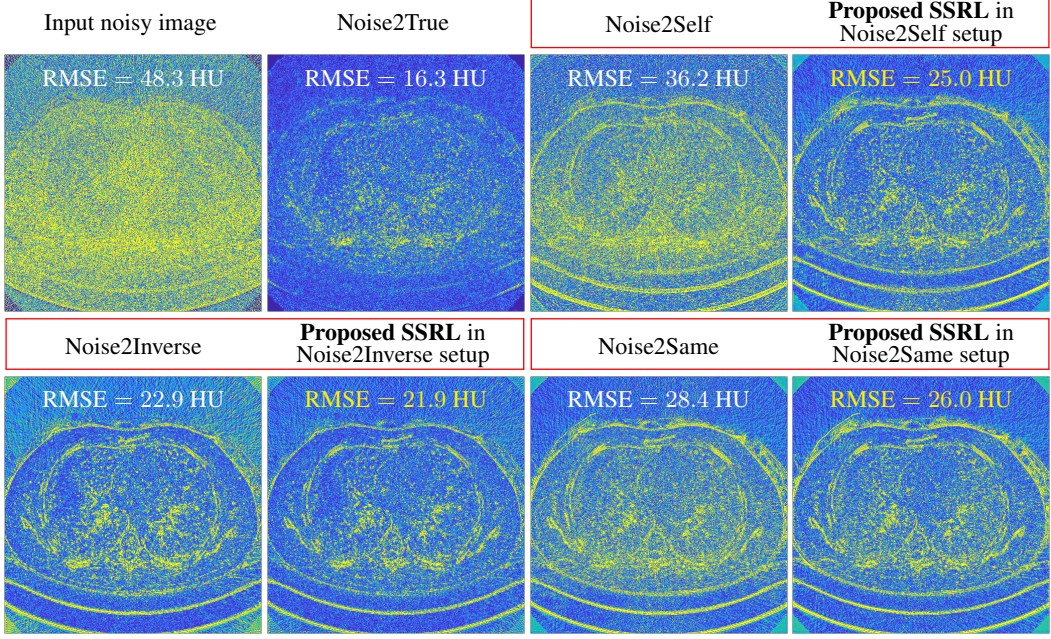

Figure S.2: Error map comparisons of denoised images via DnCNNs from different learning methods in low-dose CT (blue and yellow denote 0 and 50 absolute errors in HU, respectively). RMSE values were averaged across all test samples from The 2016 Low Dose CT Grand Challenge data.

Figures S.3 and S.4 show denoised images via U-Nets from different learning methods in camera image denoising in mixed noise and low-dose CT, respectively. These demonstrate for both applications that in all the three comparison setups, SSRL significantly improves existing self-supervised denoising methods regardless of the regression neural network architecture.

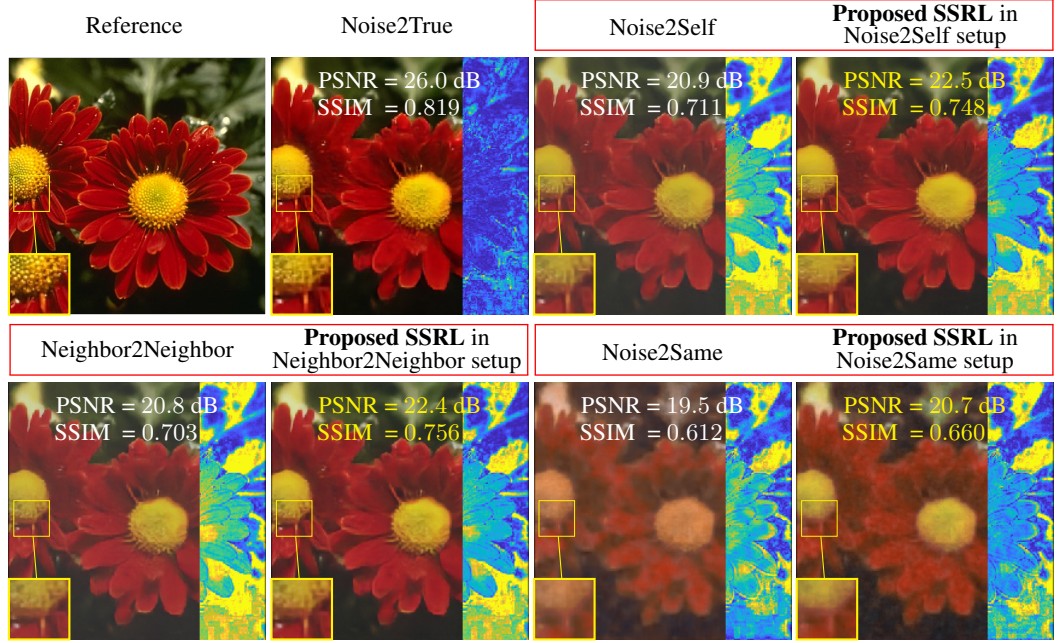

Figure S.3: Comparisons of denoised images **(left)** via U-Nets from different learning methods and their saturation error maps **(right)** in camera image denoising (blue and yellow denote 0 and 0.5 absolute errors, respectively). PSNR and SSIM values were averaged across all BSD 300 test samples.

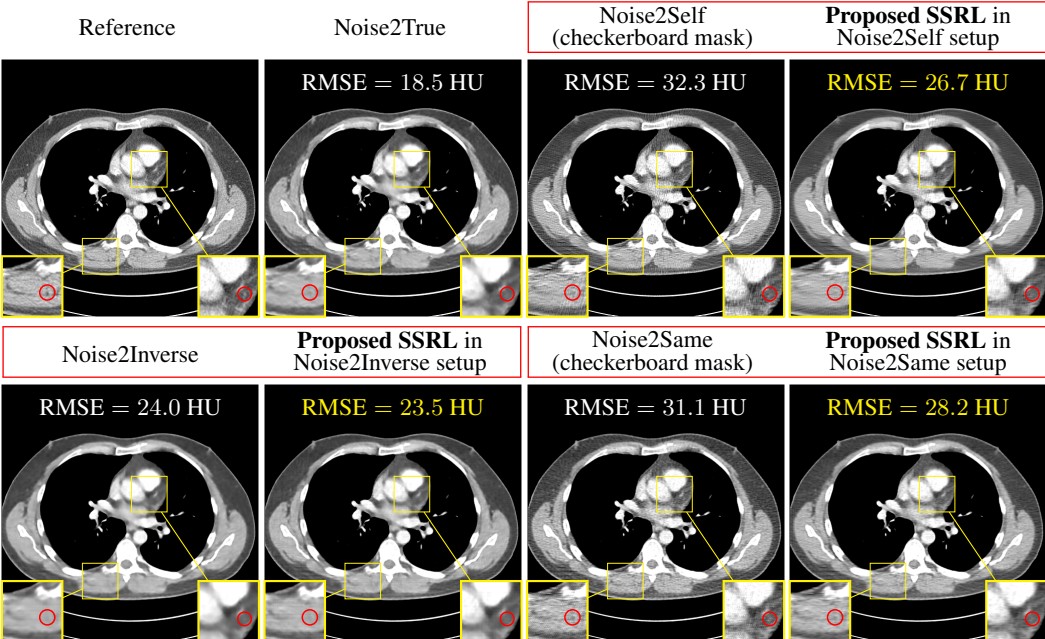

Figure S.4: Comparisons of denoised images via U-Nets from different learning methods in low-dose CT (display window is [800, 1200] HU). RMSE values were averaged across all test samples from The 2016 Low Dose CT Grand Challenge data.

Tables S.1–S.3 report quantitative image denoising results with DnCNN and U-Net with BSD 300 and Set 5 data in camera image denoising in mixed noise, and with chest slices of The 2016 Low Dose CT Grand Challenge data in low-dose CT. Red boxes in Tables S.1–S.3 compare an existing self-supervised denoising method to proposed SSRL in the corresponding setup.

Table S.1: Averaged test PSNR (dB) (**first** and **third** rows) and SSIM (**second** and **fourth** rows) comparisons with from different learning methods with DnCNN (**first** and **second** rows) and U-Net (**third** and **fourth** rows) in camera image denoising (simulated noisy *BSD 300* dataset).

| Noise2True | Noise2Self | Proposed SSRL in Noise2Self setup | Neighbor2-Neighbor | Proposed SSRL in Neighbor2-Neighbor setup | Noise2Same | Proposed SSRL in Noise2Same setup |
|---|---|---|---|---|---|---|
| 25.2 | 20.6 | **22.1** | 20.2 | **22.3** | 20.1 | **21.2** |
| 0.819 | 0.711 | **0.748** | 0.703 | **0.756** | 0.690 | **0.705** |
| 26.0 | 20.9 | **22.5** | 20.8 | **22.4** | 19.5 | **20.7** |
| 0.849 | 0.730 | **0.765** | 0.728 | **0.767** | 0.612 | **0.660** |

Table S.2: Averaged test PSNR (dB) (**first** and **third** rows) and SSIM (**second** and **fourth** rows) comparisons from different learning methods with DnCNN (**first** and **second** rows) and U-Net (**third** and **fourth** rows) in camera image denoising (simulated noisy *Set 5* dataset).

| Noise2True | Noise2Self | Proposed SSRL in Noise2Self setup | Neighbor2-Neighbor | Proposed SSRL in Neighbor2-Neighbor setup | Noise2Same | Proposed SSRL in Noise2Same setup |
|---|---|---|---|---|---|---|
| 26.4 | 19.3 | **21.2** | 19.0 | **21.9** | 18.9 | **20.0** |
| 0.890 | 0.743 | **0.785** | 0.744 | **0.806** | 0.729 | **0.753** |
| 27.2 | 19.5 | **21.8** | 19.4 | **21.6** | 17.7 | **19.3** |
| 0.910 | 0.752 | **0.802** | 0.751 | **0.800** | 0.626 | **0.712** |

Table S.3: Averaged test RMSE (HU) comparisons from different learning methods with DnCNN (**first** row) and U-Net (**second** row) in low-dose CT denoising (simulated low-dose CT dataset).

| Noise2True | Noise2Self | Proposed SSRL in Noise2Self setup | Noise2-Inverse | Proposed SSRL in Noise2-Inverse setup | Noise2Same | Proposed SSRL in Noise2Same setup |
|---|---|---|---|---|---|---|
| 16.3 | 36.2 | **25.0** | 22.9 | **21.9** | 28.4 | **26.0** |
| 18.5 | 32.3 | **26.7** | 24.0 | **23.5** | 31.1 | **28.2** |

Figure S.5 compares prediction uncertainty of trained DnCNN denoisers via different learning methods in both applications. The error bar graphs in Figure S.5 show that for both applications, in all the three comparison sets, proposed SSRL gives similar or lower prediction uncertainty over the existing self-supervised denoising methods, Noise2Self, Neighbor2Neighbor, Noise2Inverse, and Noise2Same.

Figure S.6 shows denoised camera images from SSRL-Noise2Self and SSLR-Noise2Same using the default masking parameters in Noise2Self and Noise2Same (see Section S.2.3). Compare the results in Figure S.6 with the corresponding ones in Figure 4 using the designed setups (see Section S.2.3). The comparisons demonstrate that the default and designed setups give very similar PSNR results, i.e., $\leq 0.1$ dB, but the designed setups gives slightly more visually appealing results than the default ones in Noise2Self and Noise2Same.

Figure S.7 shows denoised images from Noise2Self and Noise2Same with DnCNN and complementary checkerboard masks $J$ and $J^c$, and reports the corresponding quantitative test results, in low-dose CT denoising. Comparing the results in Figure S.7 to those of Noise2Self and Noise2Same

using the default masking setups in Figures 5 and S.2 shows that checkerboard masking improves denoising quality over default masking in Noise2Self, and achieves comparable denoising performance to default masking in Noise2Same. (See their default masking setups in Section S.2.2.) SSRL-Noise2Self with checkerboard masking achieves significantly better denoising quality compared to Noise2Self with checkerboard masking. We used pre-trained DnCNN from these two setups as $g$ in the corresponding SSRL setup.

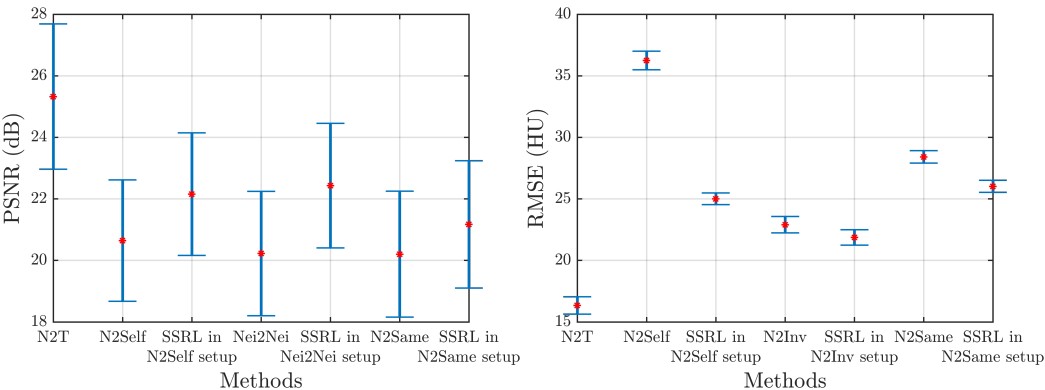

Figure S.5: Denoising performance error bars for different learning methods with DnCNN in camera image denoising in mixed noise (with BSD 300) (**left**, 300 test images) and low-dose CT (**right**, 30 test images). Red asterisks denote the averaged test PSNR or RMSE values. Error bar represents one standard deviation of test PSNR or RMSE values.

(a) SSRL-Noise2Self with the default setup in Noise2Self (i.e., $4 \times 4$ window)  (b) SSRL-Noise2Same with the default setup in Noise2Same (i.e., $14 \times 14$ window)

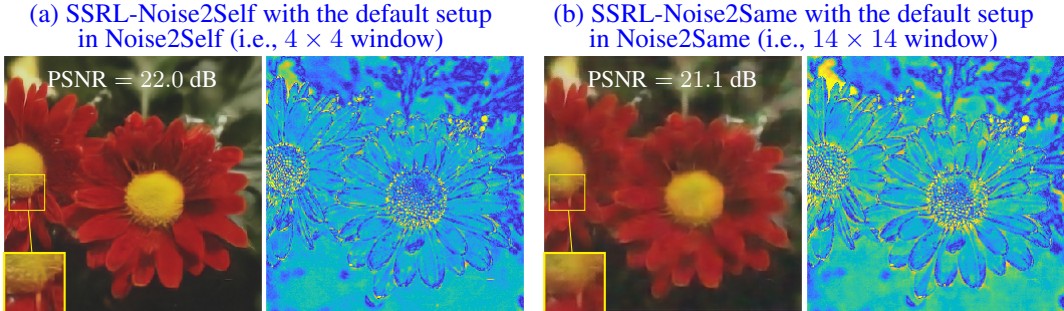

Figure S.6: Denoised images via DnCNNs and their corresponding error maps from SSRL-Noise2Self and SSRL-Noise2Same with default masks setup in camera image denoising. (In error maps, blue and yellow denote 0 and 50 absolute errors, respectively.) PSNR values were averaged across all BSD 300 test samples.

(a) Noise2Self with complementary checkerboard masks  (b) Noise2Same with complementary checkerboard masks

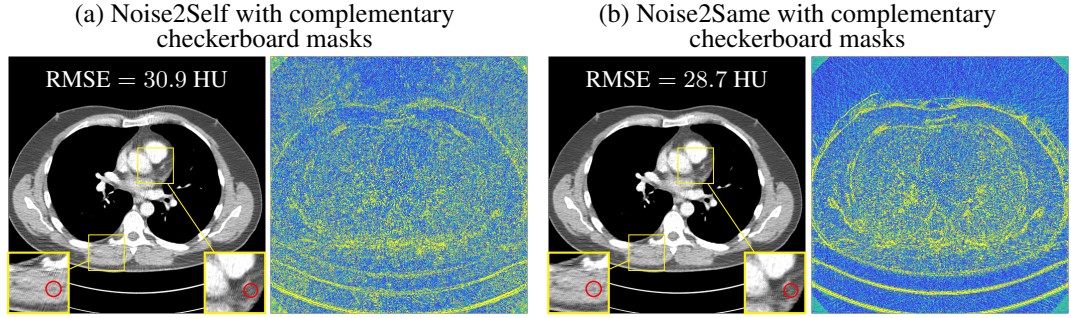

Figure S.7: Denoised images via DnCNNs and their corresponding error maps from Noise2Self and Noise2Same with complementary checkerboard masks in low-dose CT. (The display window of denoised images is $[800, 1200]$ HU; in error maps, blue and yellow denote 0 and 50 absolute errors in HU, respectively.) RMSE values were averaged across all test samples.

### S.3.2 SUPPLEMENTARY EXPERIMENTAL RESULTS WITH REAL-WORLD DATASETS

Figures S.8 and S.9 show error maps of denoised images via DnCNNs from different learning methods in camera image and low-dose CT denoising with real-world datasets, respectively. These show for both applications that in Neighbor2Neighbor, Noise2Self, or Noise2Same comparison setups, SSRL significantly improve the entire image *without* having their exact noise properties.

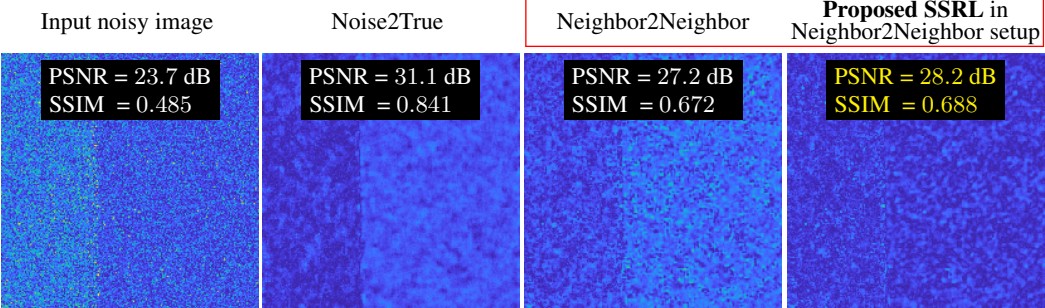

Figure S.8: Error map comparisons of denoised images via DnCNNs from different learning methods in camera image denoising with real-world dataset (blue and yellow denote 0 and 50 absolute errors, respectively). PSNR and SSIM values were averaged across all SIDD sRGB validation samples.

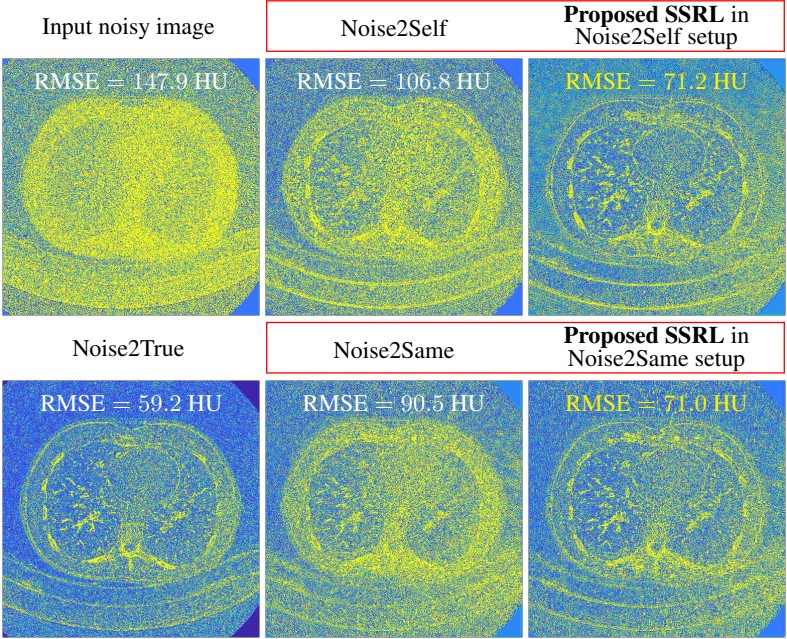

Figure S.9: Error map comparisons of denoised images via DnCNNs from different learning methods in low-dose CT with real-world dataset (blue and yellow denote 0 and 100 absolute errors in HU, respectively). RMSE values were averaged across all test samples.

### S.4 EMPIRICAL RESULTS TO SUPPORT SOME CLAIMS IN MAIN PAPER

The following empirical results support that proposed SSRL loss better approximates the supervision (Noise2True) loss than existing self-supervised learning, particularly when Assumptions 1–2 are *not* completely satisfied. We used the representative self-supervised denoising setup, Noise2Self. In the camera image denoising experiments in Section 4.1, the empirical loss values (at the last epoch) of {Noise2Self, SSRL-Noise2Self, Noise2True} are {0.296, 0.244, 0.170} (in RMSE); in the low-dose CT denoising experiments in Section 4.1, those are {2186.4, 329.1, 270.2} (in HU$^2$).

Figure S.10 empirically supports our claim in Section 3.1 that $g$ design "approximately" satisfy $\mathbb{E}[g(x)|y] = y$ in camera image denoising, with both simulated and real-word datasets. We calculated empirical $\mathbb{E}[x - y|y]$ and $\mathbb{E}[g(x) - y|y]$ with simulated noisy BSD 300 test samples (using noise model (10)) and the real-world noisy dataset (specifically, SIDD sRGB validation samples) in Section A.4.1, where $g$ is median filtering. In the simulated dataset, the empirical measures for $\{\text{avg}(|\mathbb{E}[x - y|y]|), \text{avg}(|\mathbb{E}[g(x) - y|y]|)\}$ are $\{0.0201, 0.0098\}$, where avg denotes averaging across pixels. In the real-word dataset, those are $\{0.0113, 0.0083\}$. These support our claim that $\mathbb{E}[g(x)|y] = y$ is approximately satisfied with median filtering $g$.

Figure S.11 empirically supports our claim in Section 3.2 that noise of FBP-reconstructed images in low-dose CT, i.e., $e$ in (11), has approximately zero-mean. The position of the patient table base is similar across FBP images, so it gave higher errors in the calculated sample mean; see the bottom of the image in Figure S.11.

The following empirical results support the claim in Section 3.2 that pre-trained $g$ will give better reference than $g = \mathcal{I}$ in low-dose CT denoising: the empirical measure of the Noise2Self loss (Batson & Royer, 2019) $- \mathbb{E}\|g(x_J)_{J^c} - x_{J^c}\|_2^2 -$ with simulated noisy CT test samples by setting $g$ as $\mathcal{I}$ and pre-trained DnCNN by Noise2Self are $2455.7$ and $1839.1$ (in HU$^2$), respectively.

(a) Simulated noisy dataset
(BSD 300 test samples)

(b) Real-world noisy dataset
(SIDD sRGB validation samples)

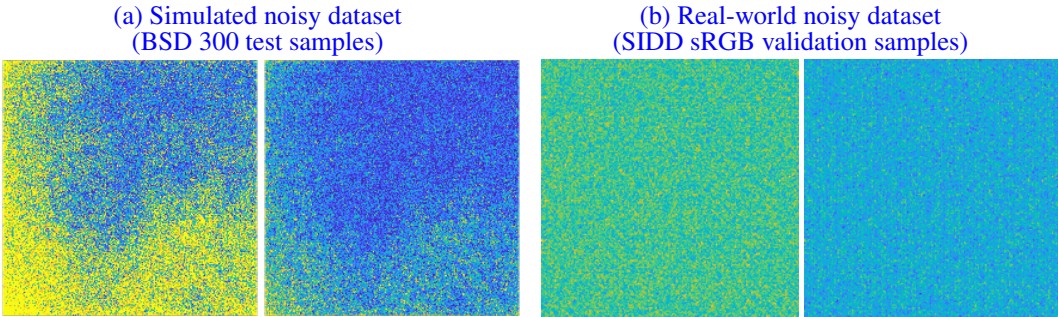

Figure S.10: Empirical observations of $|\mathbb{E}[x - y|y]|$ **(left)** and $|\mathbb{E}[g(x) - y|y]|$ **(right)** in camera image denoising ((a) Blue and yellow denote $0$ and $0.1$ absolute errors, respectively. (b) Blue and yellow denote $0$ and $0.02$ absolute errors, respectively.)

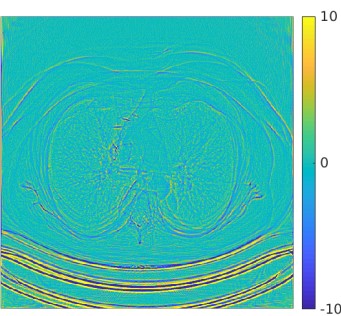

Figure S.11: Sample mean of noise in low-dose FBP images – $e$ in (11). We calculated the sample mean with 200 samples.

## S.5   EXPERIMENTAL RESULTS WITH GAUSSIAN AND POISSON+GAUSSIAN NOISE MODELS

This section studies the performance of the proposed SSRL framework with benchmark noisy datasets, MIT-Adobe FiveK data (Bychkovsky et al., 2011), corrupted by *sole* Gaussian and Poisson+Gaussian noises.

### S.5.1 EXPERIMENTAL SETUP

In Guassian denoising experiments, we simulated AWGN with the standard deviation value $\sigma_\epsilon = 25$ (Huang et al., 2021; Xu et al., 2020), and selected $g$ as Wiener filtering that is known to be optimal in the sense of minimum MSE in Gaussian denoising. In Poisson+Gaussian denoising experiments, we followed the noise simulation setup (Byun et al., 2021, Tab. 5: $(\alpha, \sigma) = (0.05, 0.02)$) that corresponds to $(\lambda, \sigma_\epsilon) = (20, 5.1)$ where $\lambda$ and $\sigma_\epsilon$ are defined in (10). To better visualize the results, we enhanced the brightness with gamma correction (with the parameter 3). We choose the representative comparison setup, Neighbor2Neighbor, from the camera image denoising experiments in Section 4.1. This is also a state-of-the-art self-supervised denoising method, particularly when only single noisy images are available (Huang et al., 2021).

### S.5.2 COMPARISONS BETWEEN DIFFERENT LEARNING METHODS

Two observations in Figure S.12. First, in both Gaussian and Poisson+Gaussian denoising, proposed SSRL using "good" $g$ further improved a state-of-the-art self-supervised denoising method, Neighbor2Neighbor. Second, in both Gaussian and Poisson+Gaussian denoising, the performance gap between Noise2True and Neighbor2Neighbor, is small, where the PSNR gap numbers well correspond to existing literature (Huang et al., 2021; Byun et al., 2021). We conjecture that this is because the noise assumptions of self-supervised denoising method are well satisfied in the aforementioned two experiments. This is connected to our conjecture in Section 4.2 that self-supervised denoising performance degrades, if its assumptions are not completely satisfied.

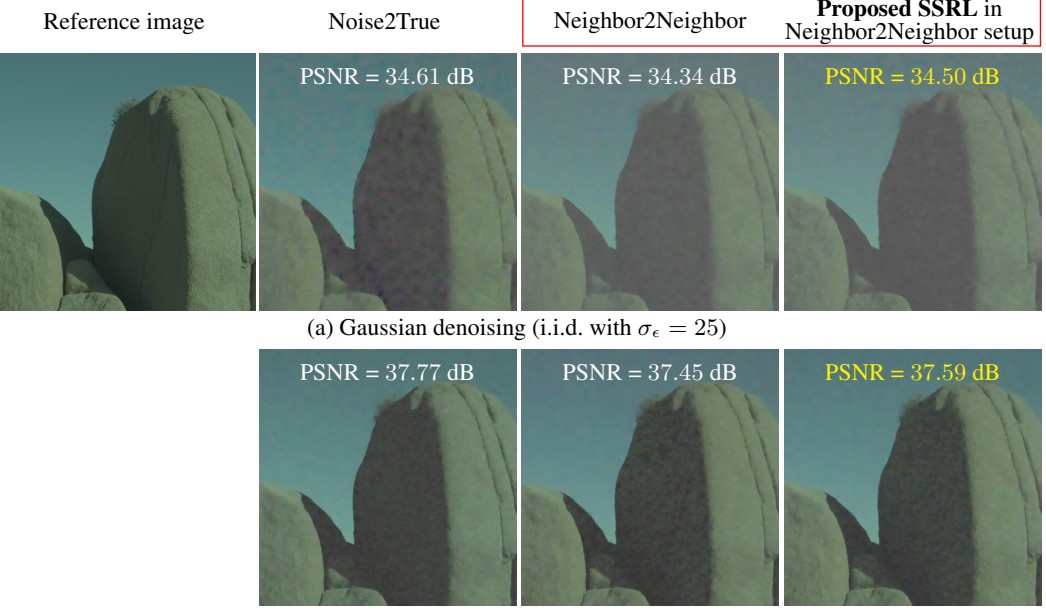

(a) Gaussian denoising (i.i.d. with $\sigma_\epsilon = 25$)

(b) Poisson+Gaussian denoising (i.i.d. with $(\lambda, \sigma_\epsilon) = (20, 5.1)$)

Figure S.12: Comparisons of denoised images via DnCNNs from different learning methods in camera image denoising. PSNR vlaues were averaged across all MIT-Adobe FiveK test samples.

## S.6 COMPARISONS TO SELF2SELF

This section compares the proposed SSRL framework with a state-of-the-art blind image denoising (a.k.a. self-supervised denoising with a single image) method, Self2Self (Quan et al., 2020), with synthetic and real-world noisy datasets for each application (see Section 4). (We used the authors' Self2Self implementation.)

First, in Figure S.13 and S.14, compare Self2Self with Neighbor2Neighbor and Noise2Self (the representative comparison setup in each simulated imaging experiment in Section 4.1), respectively. The comparisons show that Self2Self gives comparable results to Neighbor2Neighbor/Noise2Self.

Remind, however, that Self2Self is a blind denoising method, so it needs a significantly larger computations than Neighbor2Neighbor, when one denoises a new noisy image. With good $g$ designs (see Sections 3.1–3.2), proposed SSRL significantly outperformed Self2Self in both applications, regardless of whether their dataset is real or simulated.

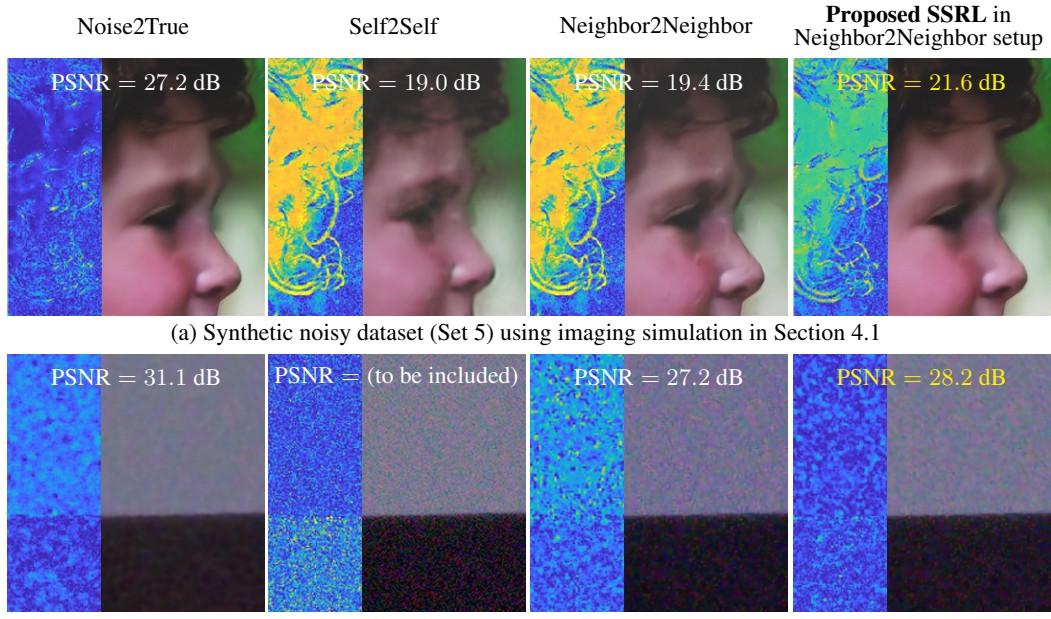

(a) Synthetic noisy dataset (Set 5) using imaging simulation in Section 4.1

(b) Real-world noisy dataset (SIDD sRGB validation)

Figure S.13: Comparisons of denoised images **(right)** from different learning methods and their error maps **(left)** in camera image denoising (blue and yellow 0 and 50 absolute errors, respectively). PSNR values were averaged across all test samples.

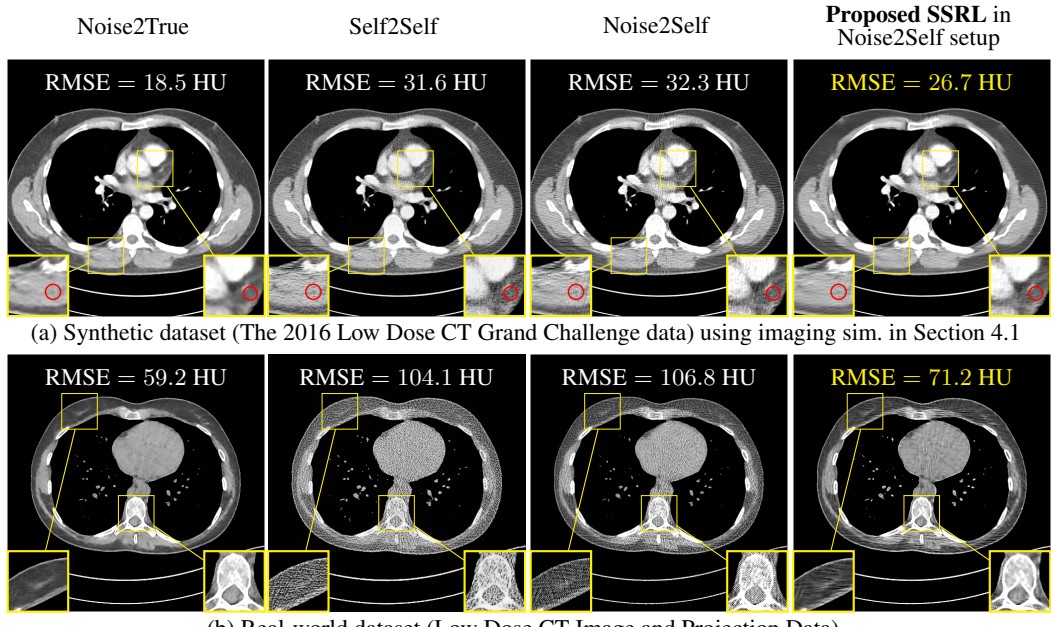

(a) Synthetic dataset (The 2016 Low Dose CT Grand Challenge data) using imaging sim. in Section 4.1

(b) Real-world dataset (Low Dose CT Image and Projection Data)

Figure S.14: Comparisons of denoised images from different learning methods in low-dose CT with synthetic and real-world datasets (display window is [800, 1200] HU). RMSE values were averaged across all test samples.

## S.7 PRELIMINARY RESULTS WITH THE TEACHER-STUDENT LEARNING PERSPECTIVE

The proposed SSRL framework is applicable to teacher-student learning (Wang & Yoon, 2021; Hinton et al., 2015; Buciluǎ et al., 2006) that aims to learn a smaller student network from bigger teacher network(s). We ran preliminary experiments in self-supervised low-dose CT denoising (using simulated data). The teacher model $g$ is the pre-trained 8-layer DnCNN by Noise2Self (with checkerboard masking), and we set the student model $f$ as $\{8, 7, 6, 5, 4, 3\}$-layer DnCNN. Applying SSRL-Noise2Self, we obtained the following numerical results: the RMSE (HU) values of student models with $\{8, 7, 6, 5, 4, 3\}$-layer DnCNNs are $\{25.0, 25.3, 25.5, 25.4, 25.2, 27.5\}$, respectively. The student DnCNNs that have the equal or lower complexity compared to its teacher network, significantly improves its teacher model of which RMSE value is 30.9 (in HU). The results might imply that student models learned from SSRL can outperform their teacher model, if they retain sufficiently high network complexity as compared to their teacher's (e.g., 3-layer DnCNN). In addition, we have additional SSRL experiment in low-dose CT denoising with the "iterative" teacher-student perspective. The teacher model $g$ is pre-trained 5-layer DnCNN from the non-iterative teacher-student SSRL method above, and we set the student model $f$ as 3-layer DnCNN. We obtained 27.3 RMSE (in HU), implying only marginal improvement over the 3-layer DnCNN obtained by the non-iterative teacher-student SSRL method. We conjecture that iterative teacher-student SSRL needs more sophisticated $g$-setups, such as an ensemble of teacher models (Hinton et al., 2015).

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
