# OpenReview forum: "Self-supervised regression learning using domain knowledge: Applications to improving self-supervised image denoising"
_ICLR.cc/2022/Conference — ICLR 2022 Submitted_

### Official Review · Reviewer_iEcr · 2021-10-20

**Correctness:** 3
**Technical Novelty And Significance:** 1
**Empirical Novelty And Significance:** 1
**Recommendation:** 3
**Confidence:** 5

**Main Review:**

*Strong points*

1. The technical writing of the paper is fairly well. The derivation can be easily  followed.

2. The paper presented a generalization of the noise2self, which conceptually can handle more tasks than noise2self.

*Weak points*

1. The novelty is limited, it is essentially a very minor generalization of noise2self, the un-bias estimator of loss function under J-invariance. There is no convincing example to show such a generalization can be realized with a constructive scheme.
2. The contribution is over-claimed. While the paper is written in the way that it seems to be able to  go beyond image denoising, it is not the case. The paper does not show how  operator $g$ can be defined for other image recovery problems with non-identity measurement matrix.
3.  Experiments are limited with very specific synthesized noise.
4.  Its practical value for self-supervised image denoising is not well justified in the experiments.  While the proposed method is a minor incremental work built on noise2self. its performance is rather well below recent existing self-denoising methods,, in terms of the performance gap to the supervised counterpart.

*Comments and Questions*

1. The paper claim a regression network which can be self-supervised. Can the author provide some examples on $g$ for other tasks that goes beyond image denoising?
2. The examples used in the experiments are removing mixture of noise. The paper did not clearly how the proposed SSRL loss function has its advantage over noise2self, in terms of its approximation to the loss function of noise2truth.
3. The proposed methods showed better performance over noise2self in the case of Gaussian+poisson+pepper-and-salt. The first 2 noises are independent, thus it fits the assumption of noise2self. The pepper-and-salt noise can be trivial treated in noise2self by excluding all pixels with value $0$ or $1$. The experiments should be conducted with such modifications on noise2self to justify its value.
4. Since noise2self, there has been a rapid progress on self-supervised image denoising which much better performance than noise2self. Indeed, the performance of recent many methods, e.g.  noiser2noise[CVPR'20], self2self [CVPR'20], Noise-as-clean [TIP'21],
Partially-linear denoiser [PAMI'21], R2R [CVPR'21], are close to the supervised counterparts. These all are based on the same idea, designing an approximate unbiased estimator to loss function.  To list some, However, From the experiments shown in this paper, the gap between the proposed one and the supervised one is not small. More experiments should be included.

**Summary Of The Paper:**

This paper present an self-supervised deep learning method for image denoising, which is a generalization of the concept proposed in noise2self. The basic idea is to approximate the unbiased estimation of the supervised loss function, which utilized the noise independence of the input and the output of the network. Some experiments are conducted on 2 image denoising tasks.


**Summary Of The Review:**

The case for this paper is quite clear to me. The paper present a minor generalization of the noise2self, which is conceptually more general, but not constructive scheme is developed to make it more applicable that noise2self. There is no concrete example to show its advantage over noise2self in terms of generalization to different problems or different noise. This paper is not a theoretical paper and its value largely lies in its empirical performance.  There have been many recent works on the unbiased estimator of the supervised loss function, the same as the proposed one. e.g.  noiser2noise[CVPR'20], self2self [CVPR'20], Noise-as-clean [TIP'21], partially-linear denoiser [PAMI'21], R2R [CVPR'21]. These methods provide competitive performance to the representative supervised denoising network, such as DnCNN, for various noise types including Gaussian, Poisson and real-world noise. The experiments conducted in this paper does not provide any comparison to these methods on standard benchmarking dataset, and the available experiments seems that the gap between the proposed one and the supervised one is quite big. In other words, the practical benefit of the proposed DDSL is not impressive.

---

> ### Author Response · Authors · 2021-11-17
> **Response to Reviewer iEcr (Part III)**
>
>
> ### Response to comment #4:
> We agree with the reviewer that recent self-supervised denoising methods can yield comparable results to the supervision method in reducing Gaussian or Poisson noise. However, there exists some performance gap in the cases where one uses more complicated noise model or stronger noise. This is supported by several recent papers [Batson \& Royer., Proc. ICML, 2019, pp. 524-533, Tab. 2], [Pang et al., Proc. IEEE/CVF CVPR, 2021, pp.2043-2052, Tab. 2], [Byun et al., Proc. IEEE/CVF CVPR, 2021, pp.5768-5777, Tab. 5]. The performance gap between self-supervised denoising and Noise2True is within the $[1.7, 3.3]$ PSNR range in dB, well-corresponding to results in the paper.
>
> Second, as described in Sections 3.1--3.2, the paper focuses on practical imaging applications that have complicated noise models with strong noise, such that existing self-supervised denoising methods could have some performance gap with a denoiser trained in a supervised manner. As described in Section 1 and (3), the paper focuses on the cases where only input samples are available. For example, the paper mainly focuses on comparisons with existing self-supervised denoising methods using noisy input samples from only a _single_ noisy realization, in particular where one does _NOT_ know noise statistic parameters. Comparing with Noisier2Noise and Noisy-As-Clean that (may) need noise statistic parameters and a blind-denoising method, Self2Self, is beyond the scope of the paper. (The paper already compares SSRL with 4 recent representative self-supervised denoising methods that can straightforwardly apply to the aforementioned scenario!) It seems that the scope of the paper is not clearly described, so we will further clarify it throughout the paper, particularly in Sections 1 and 5.
>
> Third, we ran additional experiments using a more conventional setup using the Poisson+Gaussian noise model and a benchmark dataset MIT-Adobe FiveK [Guo et al., Proc. IEEE/CVF CVPR, 2019, pp. 1712-1722], [Anwar \& Barnes, Proc. IEEE/CVF ICCV, 2019, pp. 3155-3164], [Byun et al., Proc. IEEE/CVF CVPR, 2021, pp. 5768-5777]. We followed the noise simulation setup in [Byun et al., Proc. IEEE/CVF CVPR, 2021, pp. 5768-5777, Table. 5: $( \alpha, \sigma ) = ( 0.05, 0.02)$]. The numerical results with DnCNN are given as follows (we choose the representative comparison setup from results with simulated data included in the paper): the PSNR (dB) values of \{noisy input, Noise2True, Neighbor2Neighbor, SSRL-Neighbor2Neighbor\} are \{$23.00$, $37.77$, $37.45$, $37.59$\}, respectively. Similar to the experimental results in the paper, proposed SSRL-Neighbor2Neighbor gave closer results to Noise2True, than Neighbor2Neighbor. One can also observe that the performance gap between a state-of-the-art self-supervised denoising method, Neighbor2Neighbor, and Noise2True is small, since the noise is weaker and less complicated than the simulated case in the paper. We will incorporate the new results above into Section S.2 in the supplement.
>
> Last, we would like to point out in low-dose CT denoising experiments that we used the most prevalent datasets, The 2016 Low Dose CT Grand Challenge data [McCollough, 2016] and Low Dose CT Image and Projection Data [Moen et al., 2021]; see references therein [Wang et al., IEEE Trans. Med. Imag., 37(6):1289–1296, 2018].

---

> ### Author Response · Authors · 2021-11-17
> **Response to Reviewer iEcr (Part II)**
>
>
> ### Response to comment #1:
> We present only image denoising experimental results in this paper, but the theoretical results in Section 2 hold for both the $N \neq M$ and $N = M$ cases. This is elaborated in the paper title and subtitle. We will further clarify it at the beginning of Section 2.
>
> We are preparing two separate papers that apply the proposed SSRL framework to problems beyond image denoising. For the $N \neq M$ case, we are preparing a separate paper that applies SSRL to end-to-end autonomous driving [Bojarski et al., 2016]. For the $N = M$ case where input and target are in different domain, we are preparing a separate paper that applies SSRL to image-domain dual-energy CT material decomposition [Niu et al., Med. Phys., 41(4):041901, 2014]. We cannot reveal which $g$ was used for these two applications (as our responses will be publicly available), but we used the insights included in the paper in designing/selecting $g$.
>
>  ### Response to comment #2:
> First, we would like to remind the reviewer that the paper includes experiments beyond the mixed noise case. The paper also includes experiments with Gaussian noise with zero-mean and non-identity covariance matrix (see Section 3.2) and two additionally _unknown_ noises (see Section A.4.1). Responding to Comment \#4, we will additionally include experimental results with a more conventional (but less sophisticated) noise model in camera imaging, Poisson+Gaussian. (See their results in Comment \#4.)
>
> Second, the advantage of the proposed SSRL loss function over the Noise2Self loss is demonstrated by several denoising experiment result sets with either available or unavailable noise statistics. This is supported by the theoretical result in Theorem 2 with a good $g$ design that can make $g(x)$ close to $y$. We additionally calculated empirical loss values to demonstrate that SSRL better approximates the Noise2True loss. In the natural image denoising experiments in Section 4.1, the empirical loss values (at the last epoch) of \{Noise2Self, SSRL-Noise2Self, Noise2True\} are \{$0.296$, $0.244$, $0.170$\} (in RMSE); in the low-dose CT denoising experiments in Section 4.1, those are \{$2186.4$, $329.1$, $270.2$\} (in HU$^2$). We will include these empirical observations in Section S.2 of the supplement. The paper also includes the advantage of SSRL over existing self-supervised denoising methods in terms of noise assumption relaxation. These are summarized in the paper contribution paragraph in Section 1.
>
>  ### Response to comment #3:
> We ran additional experiments as the reviewer suggested. First, we tried to exclude all pixels with $0$ and $1$ in input samples such that input to Noise2Self is $f'(x)$ instead of $x$, where $f'$ performs the above process. We found that it is very challenging to implement this for convolutions in $f$, because kernel size is fixed but the number of pixels after the exclusion process can change for different spatial locations.
>
> Next, we excluded all pixels with $0$ and $1$ in loss calculation by modifying the Noise2Self loss to $\sum_{J \in \mathcal{J}} \mathbb{E}_x || (f(x_Jc)_K - x_K||_2^2$, where $K = J \cap J_e$ and $J_e$ denotes a mask excluding pixels with $0$ or $1$ from each channel. We obtained the following numerical results with the simulated noisy BSD 300 dataset: the PSNR (dB) values of \{Noise2Self with default setup, Noise2Self with the modified loss, SSRL-Noise2Self\} are \{$20.6$, $20.5$, $22.1$\}, respectively.
>
> (please see our response to the comment #4 in Part III)

---

> ### Author Response · Authors · 2021-11-17
> **Response to Reviewer iEcr (Part I)**
>
> We thank the reviewer for constructive comments. We hope that our replies will address all concerns of the reviewer.
>
>  ### Response to weak point #1:
> **Response to concern about lack of novelty:** We agree with the review that the paper generalizes the Noise2Self loss that uses $\mathcal{J}$-invariance. However, we believe that the novelty of the paper is far beyond that. We first summarize the novelty of the proposed SSRL framework here:
> 1. The proposed SSRL framework extends far beyond the Noise2Self setup. The paper additionally applies the proposed SSRL generalization to several recent representative self-supervised denoising methods including Noise2Same [Xie et al., 2020], Noise2Inverse [Hendriksen et al., 2020], and Neighbor2Neighbor [Huang et al., 2021]. In particular, Section 2.4 discusses the SSRL approach _without_ using a generalized notion of $\mathcal{J}$-invariance (see Definition 1). With several practical imaging application examples, the paper demonstrates the outperforming performance of SSRL extensions over the aforementioned self-supervised denoising methods.
>
> 2. The proposed SSRL framework enables learning regression networks beyond denoiser learning, by using a desinable operator $g$. The paper is the first step towards self-supervised learning in regression problems, by showing that the proposed framework in Section 2 extends well to regression problems beyond image denoising. In this paper, we mainly focus on image denoising applications and are preparing separate SSRL papers for non-denoising applications; see our response to Comment \#1.
>
> 3. The paper explains how to incorporate domain knowledge into self-supervised denoising methods via $g$ and why more accurate domain knowledge can improve them. See examples in Sections 3.1--3.2. In addition, Section 3.3 proposes an empirical-loss approach for selecting $g$ if domain knowledge of specific applications is unavailable.
>
> 4. Section 2.5 shows that designable operators $g$ in SSRL can relax noise assumptions of existing self-supervised denoising methods.
> In addition, Section 4.2 includes novel experiments studying how denoising performances change with satisfying noise assumption(s) (i.e., Assumptions 1--2).
>
> To better emphasize the novelty of the paper, we will add a brief version of the above summary to Section 1.
>
> **Response to concern about lack of constructive scheme:** Domain knowledge and corresponding $g$-designs differ for different applications. The $g$-design flexibility is the key to apply the proposed framework to various regression problems, so we do _NOT_ aim to propose a concrete neural network or predictor for $g$. Having said that, Sections 2.2--2.4 specifies mathematical conditions for "good" $g$; one can use them if good domain knowledge is available. See examples in Sections 3.1--3.2. For the case where good domain knowledge is unavailable, Section 3.3 proposes an empirical-loss approach for selecting $g$. Thanks to the $g$ flexibility, we could apply the proposed method to other regression problems beyond image denoising; see our response to Comment \#1. Finally, Figure 2 demonstrates how to set $J$ and its complement for different applications; see details in Sections S.1.2--S.1.4 of the supplement.
>
>  ### Response to weak point #2:
> Please first see our response to Comment \#1. Just to clarify things, the paper also includes an imaging example with non-identity measurement matrix, low-dose CT (that collects sinogram measurement with a projection system).
>
>  ### Response to weak point #3:
> The paper also evaluates proposed SSRL with _intrinsically_ noisy natural image and low-dose CT datasets, where we do not have their complete noise properties/statistics. See Sections 4.2 and A.4.
> We will add results from a more conventional experiment setup in natural image denoising. See the third paragraph in our response to Comment \#4.
>
>  ### Response to weak point #4:
> The proposed method extends beyond Noise2Self; see our response to Weak Point \#1. Regarding the performance gap between the self-supervised and supervised methods, please see our response to Comment \#4.
>
> (please see our responses to the Comments in Part II and III)

---

> ### Comment · Reviewer_iEcr · 2021-11-18
> **Comments on the response**
>
> 1. The claim of the applicability of the proposed method for removing unknown noise or mixture noise while others cannot  is  not theoretically justified. From empirical evaluation, such a claim is not well supported either. Many methods can deal with real-world noise in practice well. It is just for theoretical analysis, certain assumptions needs to be imposed. From this viewpoint, the claim is convincing only if solid experiments are conducted over real-world data. It is hard to convince people the effectiveness of the proposed method for processing real-world data while all experiments are  synthesized noise, not real-world noise.
>
> 2. Indeed, many existing works I listed, e.g. noiser2noise, self2self, partially-linear denoiser, R2R They have the experiments conducted on real-world data, not just Gaussian white noise. It is true that  their mathematical derivation is based on Gaussian white noise, but it does not mean they are not effective for deal with other noise. Indeed,  their performance seems to be close to the supervised method in their experiments. In contrast, the proposed method's performance is far behind of that of the supervised method in the experiments. While the benchmarking dataset is different, the results cannot support the effectiveness of the proposed method on dealing with complex noise or real-world noise, and the proposed one did not do well on Gaussian noise.
>
> 2. The authors mentioned they have another paper on using the proposed one for inverse problem. I don't see how it is related to this paper. The works presented here cannot be directly used for inverse problems, as $y$ in the inverse problem only provides measurement in the range space of measuring matrix. It is not straightforward to see how ambiguity in null space can be handled by the proposed one from Theorems. If its applicability is based on empirical results. Then, it is a different story, every method can try it empirically. It is just a matter of performance.

---

> > ### Author Response · Authors · 2021-11-21
> > **Response to Reviewer iEcr on Nov. 18 (Part III)**
> >
> > ### Response to Comment \#3:
> > It may not be straightforward to see how ambiguity in null space can be handled by the proposed SSRL methods in Theorems.
> > Suppose that an implicit function $g$ (approximately) satisfies Assumption 3 and gives a unique solution. We conjecture that using SSRL with such $g$ learns a regression net $f$ in a self-supervised way that implicitly handles the null space of an inverse problem. In low-dose CT reconstruction, for example, $x$ is a noisy sinogram and $g$ could be set as some regularized least-squares solver that provides a unique solution. By setting $f$'s architecture to a regression NN architecture from recent works like AUTOMAP [Zhuet et al., Nature, 555:487–492, 2018], proposed SSRL could learn $f$ that directly maps noisy sinograms to good quality images while avoiding the non-uniqueness issue, in a self-supervised manner. (These are similarly applied to our dual-energy CT decomposition research that was mentioned in our response to initial Comment \#1.) Alternative approaches include adding regularization perspectives to $f$, e.g., null-space learning [Schwab et al., Inv. Prob., 35(2):025008, 2019] and iterative neural network [Adler \& Öktem, Inv. Prob., 33(12):124007, 2017]. To avoid any potential related misunderstanding, we will add a short comment that applying SSRL to other regressions problems may need careful investigations about pseudo-predictor $g$ based on their domain knowledge. (Again, solving inverse problems by learning one-step regression NNs that map from one domain to another different is beyond the scope of the paper. This is emphasized in the paper subtitle and clarified in several places in the paper.)
> >
> > To the best of our recollection, the paper and our responses did/do **NOT** argue or imply anywhere that one can "straightforwardly" -- i.e., _without_ any further careful investigations -- apply SSRL to solving any inverse problems. If the reviewer can find any such implications in the paper, could you point them out? We will happily correct them (if exist).
> >
> > ### Contribution of the paper:
> > Our theoretical understanding and empirical denoising results with **several** real-world and simulated datasets with **diverse** noise properties consistently show the following message:
> >
> > _The proposed SSRL framework can significantly improve many existing self-supervised denoising methods, by using a good pseudo-predictor $g$ that leads its prediction close to ground-truth and/or substantially relaxes existing self-supervised denoising assumptions._
> >
> > Our work is the first to "theoretically understand" that under some relaxed conditions, self-supervised student learning using $g$ as a teacher can become equivalent to using ground truth to train a new regressor $f$. Such theoretical understandings provide guidelines to design good $g$ that can improve the performance of SSRL. We ran extensive experiments to resolve all concerns from the reviewers, and including them into the paper will further improve the contribution of the paper. We strongly believe that our contributions are significant and it is worth presenting them to the audience at ICLR for further SSRL investigations and its potential applications.

---

> > ### Author Response · Authors · 2021-11-21
> > **Response to Reviewer iEcr on Nov. 18 (Part I)**
> >
> > Thank you for your quick reply to our responses. Even though we have some disagreements, we still appreciate your valuable time and comments. We ran extensive experiments, hoping that those will address all concerns of the reviewer.
> >
> > ### Response to Comment \#1 and partial Comment \#2 concerning no experiments with real-world datasets:
> > The intrinsically noisy datasets used in the paper's experiments **ARE** real-world datasets that were collected by real-world camera [Abdelhamed et al., 2018] and real-world CT scanner with patients [Moen et al., 2021]. (In particular, [Moen et al., 2021] is the most widely used benchmarking dataset in CT image denoising and reconstruction research.) The paper includes related materials at the beginning of Section 4, and in Section 4.2: Camera image denoising and low-dose CT denoising with real-world datasets and Section A.4 in the appendix; see also our response to Weak Point \#3. The theoretical understandings in Section 2 well correspond to the empirical observations in Section 4 with **two** _real-world_ noisy datasets, similar to what we found with **three** simulated imaging experiment sets (including the one to resolve initial Comment \#4). To avoid this potential confusion, we will refine the wording "intrinsically noisy datasets" with "real-world datasets," throughout the paper, appendix, and supplement.

---

> ### Author Response · Authors · 2021-11-23
> **Major changes made in the paper**
>
> This comment lists changes made in the revised paper, appendix, and supplement to resolve the reviewer's concerns (changes are in blue):
> + Responding to Weak Point \#1, we further elaborated the paper contributions in Section S.1 of the supplement.
> + To resolve initial Comment \#2, we added a empirical observation in Section S.4 (first paragraph) of the supplement.
> + Responding to initial Comment \#4, we first clarified the scope of the paper in Section 1, and at the beginning of Sections 3 \& 4. To resolve the gap inconsistency issue compared to existing results (in Comment \#4), we added Gaussian and Poisson+Gaussian denoising experimental results in Section S.5 of the supplement.
> + To avoid the confusion about the wording "intrinsically noisy dataset" (see 11/18 Comment \#1), we changed it to "real-world dataset" throughout the paper, appendix, and supplement.
> + To resolve 11/18 Comment \#2, we added Gaussian and Poisson+Gaussin denoising experimental results and Self2Self results with _four_ real or synthetic noisy datasets in Sections S.5 and S.6 of the supplement, respectively.
> + To avoid potential misunderstanding that proposed SSRL can be straightforwardly applied to any inverse problems (see 11/18 Comment \#3), we added a clarification to Section 5.

---

### Official Review · Reviewer_1ABd · 2021-11-01

**Correctness:** 3
**Technical Novelty And Significance:** 3
**Empirical Novelty And Significance:** 3
**Recommendation:** 6
**Confidence:** 2

**Main Review:**

1. Examples in Motivation are not convincing. There is no doubt that ground-truth clean image can provide better denoising performance, and any other modifications will yield denoising performance decreases.

2. If I understand right, when training denoiser f, the supervision is from complementary denoising results by pre-designed g. The masks on J and Jc have no intersection. From Fig. 2, it is confused to understand the training procedure. Predicted output in above should be masked on J? Why pseduo-target is entire image while the one in bottom is masked? Also in experiments, the results by unbalanced masks are not given, so why give this setting in Fig. 2?

3. The proposed method is evaluated on their designed settings, which is not consistent with most exisitng self-supervised denoising methods.

**Summary Of The Paper:**

This paper proposed a self-supervised denoing method using domain knowledge. The proposed method seems somewhat like knowledge distillation, where a pre-trained denoiser or handcrafted designed noise models g can provide better initial results than original noisy images, and then a better denoiser f can be trained based on the results of g instead of noisy images. By forcing the complement of f and g, f will not be same with g, and the denoising performance is bossted.

**Summary Of The Review:**

This work proposed an interesing self-supervised denoising method and its improvements are significant. But it has several issues to be addressed listed above.

---

> ### Author Response · Authors · 2021-11-17
> **Response to Reviewer 1ABd**
>
> We thank the reviewer for careful reviews and valuable comments. We hope that our replies will address all concerns of the reviewer.
>
> ### Response to comment #1:
> We will use alternative examples to better demonstrate that $g$ with good domain knowledge improves the denoising performance of $f$. We ran denoising experiments with natural images corrupted by salt-and-pepper noise, by choosing median filtering and BM3D denoiser [Mäkinen et al., IEEE Trans. Image Process., 29:8339-8354, 2020] for $g$ in (3). The PSNR (dB) values for the former and latter $g$-setups are \{$25.8$, $23.1$\}. One knows that median filtering is effective for reducing salt-and-pepper noise, and the examples above would well demonstrate that good domain knowledge improves the denoising performance of $f$.
>
> ### Response to comment #2:
> The reviewer's understanding is correct. In Figure 2 (top), $f$ and $g$ use almost equal amount of information, so they can predict the entire image (and masking was not needed). In Figure 2 (bottom), since $g$ uses much less information than $f$, it is challenging for $g$ to predict the entire image. See Section S.1.4. We will further clarify this in Section 2.3. Furthermore, Section S.1.3 in the supplement specified that we used the scheme in Figure 2 (bottom) for natural image denoising experiments.
>
> ### Response to comment #3:
> Similar to existing self-supervised denoising methods, masking parameters are tunable in SSRL. In natural image denoising experiments using median filtering (within the SSRL framework), the default (e.g., $4 \times 4$ window in Noise2Self) and designed (e.g., $3 \times 3$ window in SSRL-Noise2Self) masking parameters showed very similar PSNR results, i.e., $\leq 0.1$ dB, but the latter parameters gave slightly better visual results. We will include results with the default masking parameters in Section S.2 of the supplement. (In experiments with the existing self-supervised denoising methods, Noise2Self and Noise2Same, the both setups gave very similar results both qualitatively and quantitatively.)
>
> We observed in low-dose CT denoising experiments using pre-trained $g$ that using the default setup, i.e., $| J | \ll | J^c |$, gives poor prediction in both SSRL-Noise2Self and -Noise2Same. (See related discussion in Section S.1.4.) We thus used complementary checkerboard masks for those setups. The paper also reports that in Noise2Self and Noise2Same, complementary checkerboard masks gave better than or comparable results with the default donut-filter setup; see Figure S.6-related paragraphs in Section S.2 of the supplement.

---

> ### Author Response · Authors · 2021-11-23
> **Major changes made in the paper**
>
> This comment lists changes made in the revised paper, appendix, and supplement to resolve the reviewer's concerns (changes are in blue):
>
> + To resolve Comment \#1, we replaced examples with more convincing ones in Section 2.1.
>
> + To resolve Comment \#2, we added a clarification in Section 2.3.
>
> + To resolve Comment \#3, we added a clarification and experimental results in Sections S.2.3 and S.3.1 of the supplement, respectively.

---

### Official Review · Reviewer_vDqQ · 2021-11-02

**Correctness:** 4
**Technical Novelty And Significance:** 3
**Empirical Novelty And Significance:** 3
**Recommendation:** 6
**Confidence:** 3

**Main Review:**

I think it is a good paper to improve the performance of existing unsupervised denoising methods. However, I have several concerns:
1. One statement of this paper is the domain knowledge is important which can further improve the performance of unsupervised denoising. This is based on that Eq. 3 is closer to Eq. 1 if the 2nd term of Eq. 5 is close to zero. However, the second term of Eq. 5 is not relevant to f(x). In this way, is the back propagation of Eq. 3 and Eq. 1 the same when optimizing f?
2. A g(x) closer to the ground truth can lead to a better result for f(x) according to my understanding. In this way, g(x) can also be treated as a denoiser. Can the authors explain why f(x) has a better denoising ability than g(x)?
3. The visual improvement is too marginal even though the PSNR is higher e.g. in Fig. 4.
4. It seems spatially correlated noises may violate the assumptions in this paper. The authors provide some experiments to show the effectiveness of the proposed method in real image denoising dataset (Abdelhamed et al., 2018). Can the authors provide more analysis about why the proposed method still works well for spatially correlated noises?
5. The motivation is strange to me in Sec. 2.1. It is strange to use norm(y) as supervision in Fig.1 (right).
6. In sec. 3.1 and 3.2, Fig. A should be revised to Fig. 3?

**Summary Of The Paper:**

This paper proposes a method for unsupervised image denoising. It shows that a better designed operator based on domain knowledge can benefit unsupervised image denoising task. The provided experimental results show the proposed method outperforms existing unsupervised denoising ones and achieves similar performance to supervised methods.

**Summary Of The Review:**

This paper proposes a novel method to improve the performance of unsupervised denoising methods. But I still hope the authors address my concerns above.

---

> ### Author Response · Authors · 2021-11-17
> **Response to Reviewer vDqQ**
>
> We thank the reviewer for careful reviews and valuable comments. We hope that our replies will address all concerns of the reviewer.
>
> ### Response to comment #1:
> Yes, the reviewer's understanding is correct, under the setup given in Theorem 2. In this project, we mainly focus on practical imaging applications that have complicated noise models, such that Assumptions 1--2 are not perfectly satisfied. To overcome such limitations, we proposed the SSRL framework and aim to design good $g$ such that $g(x)$ becomes close to $y$. It seems that the scope of the paper is not clearly described, so we will better clarify this throughout the paper.
>
> While preparing our response to the reviewer's comment, we realized that motivation to design good $g$ in Section 2.3 is misleading.
> To resolve this issue, we will first add a new theoretical result to Theorem 2 that optimal $f$ for (5) is given by $f^\star (x_{J^c}) = \mathbb{E}[ g(x_J) | x_{J^c} ] = \mathbb{E}[ y | x_{J^c} ]$. If the given assumptions (in Section 2.1) are not perfectly satisfied like in four examples in the paper, the second equality will not hold. This motivates to design good $g$ such that $g(x_J)$ is close to $y$ and thus, self-supervised optimal solution $\mathbb{E}[ g(x_J) | x_{J^c} ]$ becomes closer to supervised optimal solution $\mathbb{E}[ y | x_{J^c} ]$. We will elaborate this in Section 2.3 and make appropriate changes throughout the paper.
>
> Speaking with the perspective of backpropagation, if given assumptions are not perfectly satisfied, the inner product term in (4) may back-propagate residual and undesired errors to the network, thus can potentially induce the gap between (3) and (1).
>
> ### Response to comment #2:
> The reviewer's understanding is correct, and we would like to first elaborate the regime of $g$ considered in the paper. The paper mainly considers in image denoising that $g$ is either pre-trained from existing self-supervised denoising methods or hand-crafted (see the paragraph below Theorem 2), such that $g$ has some performance gap with a denoiser trained in a supervised way. This is natural/reasonable for practical imaging applications mentioned at the first paragraph in our response to Comment \#1; see, also, Section 3. In such cases, we think that having better pseudo-target $g(x)$ in SSRL can lead to better $f$ over $g$ -- that is trained via existing self-supervised denoising methods using no pseudo-predictor. If one has a very good denoiser such that $g(x) \approx y$, then one does not need to use SSRL to train another $f$. We will further clarify this at the beginning of Section 2.
>
> ### Response to comment #3:
> We think that the PSNR gains are from the combination of better color saturation and detail preservation. Color saturation preservation is particularly important as it affects the overall composition and mood of pictures. To better see saturation differences, one may need a little different monitor setup that gives good color accuracy. To better emphasize saturation preservation differences, we included saturation error maps in Figure 4.
>
> ### Response to comment #4:
> First, we would like to remind that both the proposed SSRL method and its specialized case, e.g., Noise2Self, still work, even if Assumption~1 is not satisfied (e.g., pixel-wise correlated noise). If it is satisfied, their performances will improve. This is discussed at the last paragraph in Section 4.2: Low-dose CT denoising (simulated data).
>
> Our understanding why SSRL works for the specified real image denoising dataset relates to the selected $g$ design, median filtering, that "approximately" satisfy Assumption 2. We calculated empirical $\mathbb{E}[g(x)-y|y]$ and $\mathbb{E}[x-y|y]$ with the sepcified dataset. The empirical values for the former and latter measure are  are $\mathbf{0.0083}$ and $0.0113$, respectively, supporting the claim that Assumption 2 ($\mathbb{E}[g(x)|y] = y$) is approximately satisfied with median filtering $g$. This is similar to our empirical findings with the synthesized counterpart dataset; see Figure. S.7-related discussion in Section S.2 of the supplement. We will include the aforementioned new empirical observations to Section S.3 in the supplement.
>
> ### Response to comment #5:
> We will use alternative examples to better demonstrate that $g$ with good domain knowledge improves the denoising performance of $f$. We ran denoising experiments with natural images corrupted by salt-and-pepper noise, by choosing median filtering and BM3D denoiser [Mäkinen et al., IEEE Trans. Image Process., 29:8339-8354, 2020] for $g$ in (3). The PSNR (dB) values for the former and latter $g$-setups are \{$25.8$, $23.1$\}. One knows that median filtering is effective for reducing salt-and-pepper noise, and the examples above would well demonstrate that good domain knowledge improves the denoising performance of $f$.
>
> ### Response to comment #6:
> Oh, yes, the reviewer is correct. We will change Figure A.1 to Figure 3 in Sections 3.1--3.2.

---

> ### Author Response · Authors · 2021-11-23
> **Major changes made in the paper**
>
> This comment lists the changes in the revised paper, appendix, and supplement that were made to resolve the reviewer's concerns (changes are in blue):
>
> + To resolve Comment \#1, we refined our motivation to design good $g$ in Sections 2.3 (Theorem 2 and below), 2.4 (below Theorem 3), 3.3 and 5. We also clarified the scope of the paper at the beginning of Section 3.
>
> + To resolve Comment \#2, we added a clarification at the beginning of Section 2.
>
> + To resolve Comment \#4, we added a corresponding analysis in Section S.4 (second paragraph) of the supplement.
>
> + To resolve Comment \#5, we replaced examples with more convincing ones in Section 2.1.

---

### Decision · Program_Chairs · 2022-01-20

**Decision:**

Reject

**Comment:**

This paper focuses on unsupervised image denoising and proposes a method to do so. It shows that using a designed operator based on domain knowledge can help improve unsupervised image denoising. The authors also provide experimental results demonstrating that the proposed methods outperform existing unsupervised denoising and behave similar in performance to supervised methods. The reviewers liked the improvements but (1) limited novelty/simple extension of noise2self, (2) example not convincing, (3) lack of clarity in 2.3, (4) a variety of other technical concerns. The authors partially addressed these concerns. However, I concur with the reviewers that the paper still requires more work and is not ready for publication in its current form.